# Hillslope denudation and morphologic response to a rock uplift gradient

Vincent Godard[1,2], Jean-Claude Hippolyte[1], Edward Cushing[3], Nicolas Espurt[1], Jules Fleury[1], Olivier Bellier[1], Vincent Ollivier[4], and ASTER Team[*1]

[1]Aix-Marseille Univ, CNRS, IRD, INRA, Coll France, CEREGE, Aix-en-Provence, France
[2]Institut Universitaire de France (IUF)
[3]PSE-ENV/SCAN, Institut de Radioprotection et de Sûreté Nucléaire, Fontenay-aux-Roses, France
[4]Aix-Marseille Univ, CNRS, Minist Culture, LAMPEA, Aix-en-Provence, France

**Correspondence:** Vincent Godard (godard@cerege.fr)

**Abstract.** Documenting the spatial variability of tectonic processes from topography is routinely undertaken through the analysis of river profiles, since a direct relationship between fluvial gradient and rock uplift has been identified by incision models. Similarly, theoretical formulations of hillslope profiles predict a strong dependence to their base level lowering rate, which in most situations is set by channel incision. However, the reduced sensitivity of near-threshold hillslopes and the limited availability of high-resolution topographic data has often been a major limitation for their use to investigate tectonic gradients. Here we combined high-resolution analysis of hillslope morphology and cosmogenic nuclide-derived denudation rates to unravel the distribution of rock uplift across a blind thrust system at the Southwestern Alpine front in France. Our study is located in the Mio-Pliocene Valensole molassic basin, where a series of folds and thrusts has deformed a plateau surface. We focused on a series of catchments aligned perpendicular to the main structures. Using a 1-m LiDAR Digital Terrain Model, we extracted hillslope topographic properties such as hilltop curvature $C_{HT}$ and non-dimensional erosion rates $E^*$. We observed systematic variation of these metrics coincident with the location of a major underlying thrust system identified by seismic surveys. Using a simple deformation model, the inversion of the $E^*$ pattern allows us to propose a location and dip for a blind thrust, which are consistent with available geological and geophysical data. We also sampled clasts from eroding conglomerate at several hilltop locations for $^{10}$Be and $^{26}$Al concentration measurements. Calculated hilltop denudation rates range from 40 to 120 mm/ka. These denudation rates appear to be correlated with $E^*$ and $C_{HT}$ extracted from the morphological analysis, and are used to derive absolute estimates for the fault slip rate. This high-resolution hillslope analysis allows us to resolve short-wavelength variations in rock uplift that would not be possible to unravel using commonly used channel profile-based methods. Our joint analysis of topography and geochronological data supports the interpretation of active thrusting at the Southwestern alpine front, and such approaches may bring crucial complementary constraints to morphotectonic analysis for the study of slowly slipping faults.

# 1    Introduction

The topography of the Earth evolves in response to surface processes driven by external forcing of tectonic and climatic origins (e.g. Champagnac et al., 2012; Whittaker, 2012). Information about tectonic uplift is sequentially transmitted through landscapes, first by the adjustment of river gradients, which then set the local base level of hillslopes. The water supply and temperature set by climatic conditions control the efficiency of weathering, erosion and transport processes across the Earth surface. The present-day land surface morphology results from this accumulated actions of tectonic and climatic forcing through time, and a major endeavor of geomorphological research is to interpret measurable topographic properties in terms of space and time variations of either these tectonic uplift or climatic conditions (e.g. Roberts et al., 2012; Demoulin, 2012; Fox et al., 2014). In particular, documenting the spatial variability of tectonic processes from topographic analysis has been a key research focus (Wobus et al., 2006), as changes in topographic gradients could record variations in rock uplift rates at various scales, from regional patterns associated with crustal or lithospheric deformation (e.g. Gallen et al., 2013), down to differential motion across individual faults (e.g. Boulton et al., 2014). Such investigations have often been motivated by the practical concern of identifying high strain zones in tectonically active regions in order to contribute to seismic hazard assessment (e.g. Morell et al., 2015).

These studies have been, by a large margin, dominated by approaches relying on the analysis of river long-profile properties, as a direct relationship between fluvial gradient and rock uplift had been identified in many incision models (Tucker and Whipple, 2002). Notably, the computation of morphometric parameters such as steepness indexes, a measure of channel gradient normalized for drainage area (Kirby and Whipple, 2012), is now a standard approach when investigating river networks, as this parameter is theoretically dependent on rock uplift and has been shown to be positively correlated with field measurements (e.g. Duvall et al., 2004; Wobus et al., 2006; Cyr et al., 2010). The great development of these methods has been made possible by the increasing availability of medium resolution Digital Elevation Models (DEM : 10 to 100 m pixel size), which allow the reliable extraction of river profiles and accurate computation of along-stream gradients. While robust, efficient and widely used in a variety of settings, these fluvial-based approaches can encounter important limitations when dealing with complexities of river systems such as lithological variations, transient evolution, or along-stream changes in fluvial dynamics, but are also inherently constrained by the planform distribution of river networks, which might not always be optimal to sample the rock uplift patterns.

Similarly to river profiles, theoretical formulations of hillslope denudation predict a strong dependence of morphological parameters, such as slope or relief, on the rate of base level fall set by channel incision. However, key elements of hillslope behavior such as the threshold stability angle for hillslope material and the non-linear relationship between sediment fluxes and topographic gradient (Roering et al., 1999, 2001, 2007), implies that for fixed valley spacing, hillslope morphology can be insensitive to changes in erosion or rock uplift rates over a large range of values (Burbank et al., 1996; Ouimet et al., 2009). This behavior is a major limitation for the use of hillslope morphology to retrieve information about tectonic gradients. Furthermore, from a methodological point of view, the widely available intermediate resolution DEM, which are appropriate to describe river profiles developing over 1-100 km length scales, will only provide a very coarse description of hillslopes with

typical lengths of the order of 100 m (Grieve et al., 2016a).

Nevertheless, this methodological limitation is currently being overcome by the growing use of LiDAR or photogrammetric techniques delivering Digital Terrain Models (DTM) with sub-metric resolutions (James and Robson, 2012; Glennie et al., 2013; Passalacqua et al., 2015). The widespread distribution of such data has spurred a great interest in new approaches to ex-
tract relevant information at the hillslope scale (e.g. DiBiase et al., 2012; Hurst et al., 2012; Grieve et al., 2016b, c; Milodowski et al., 2015; Clubb et al., 2020). It is noteworthy that the high-resolution of these DTMs allows to compute accurate derivatives of the topographic surface. According to linear diffusion theory, the second derivative, or curvature, covaries with erosion rate. This relationship remains valid for near-threshold hillslopes in the vicinity of the hilltop where topographic gradients are usually small (Hurst et al., 2012). This possibility to access reliable proxies for erosion rate from hillslope scale metrics has
lead to reconsider the potential of hillslope analysis for the assessment of denudation gradients, which can be used to infer patterns of uplift. Notably, Roering et al. (2007) have proposed a conceptual framework, based on a formulation for non-linear hillslope sediment flux, which highlights the links between steady-state hillslope morphology and the dynamics of erosion processes as well as the underlying tectonic or climatic forcings. Hurst et al. (2012) have built upon this formulation to construct a joint analysis of high-resolution topographic hillslope metrics and Cosmogenic Radionuclides (CRN) data in the Sierra
Nevada (California). In particular, they used CRN-derived denudation rates to calibrate the efficiency parameter for hillslope transport processes, and constrain the distribution of absolute erosion rates from hilltop curvature measurements. Similarly, Hurst et al. (2013a) were able to finely track transient landscape adjustment along the San Andreas Fault where long-term motion is progressively moving hillslopes in and out of a high-uplift-rate pressure ridge. This localised change in the tectonic boundary condition is closely recorded by hillslope relief or slope angle and hilltop curvature extracted from LiDAR data, with
the growth and decay phases of landscape evolution leaving a distinctive signature.

These promising results have offered hillslope-based critical insights into the dynamics of transient landscapes, with a spatial density of information several orders of magnitude higher than what could be resolved with approaches based on the fluvial network. Important methodological developments were necessary to extract the relevant information from high-resolution DTMs (e.g. Grieve et al., 2016b). However, while theoretically warranted, the applicability of these approaches to explore tectonic
gradients has only been tested on a limited number of cases (e.g. Hurst et al., 2013a, 2019; Clubb et al., 2020). There is therefore an urgent need to further investigate hillslope morphological response in various types of tectonic settings to unravel the potential of such methods as an alternative or complement to the routinely applied investigation of river profiles. Additionally, even at high resolution, the analysis of hillslope or river morphological properties can only deliver estimates of the relative intensity of surface processes, the actual rates and efficiency of which can only be obtained through geochronological techniques
such as cosmogenic nuclides. When facing the growing availability of such data there is a critical need to assess whether and under which circumstances the high-resolution topographic properties of landscapes and their rates of evolution can both be framed into a coherent picture on the basis of available theoretical formulations for landscape evolution (Dietrich et al., 2003). The objective of our study is to investigate the changes in hillslope morphology, as observed with a LiDAR DTM, across a rock uplift gradient at the front of the Southwestern Alps, France, and to assess what kind of information can be retrieved concerning
the underlying tectonic processes. We also confront this spatial structure of the landscape with denudation rates derived from

cosmogenic nuclides in order to compare the relative spatial distribution of surface processes and uplift rate inferred from the DTM analysis with absolute values. In the following, we first present the main features of our working area, the Puimichel Plateau in the Mio-Pliocene Digne-Valensole basin, with a focus on key aspects of its main structures and history that make it an interesting place to investigate the interactions between tectonic forcings and surface processes response. Then, we introduce the morphological analysis and cosmogenic nuclide methods we used, and we describe the corresponding datasets produced during this study. Finally, we discuss the implications of these results in terms of the imprint of tectonic gradients into hillslope morphology, the constraints that can be put on tectonic structures at depth from high-resolution topographic data, and their relationship with denudation rates calculated from cosmogenic nuclide concentrations.

## 2 Setting

The studied area is located in the southern part of the Western European Alps, where mountain ranges result from a combination of Pyrenean (Late Cretaceous to Eocene) and Alpine (Neogene) tectonic phases (Figure 1). However, the present-day tectonic activity in the area is considered to be low, and there is no significant horizontal strain rate resolved from geodetic data, despite the occurrence of earthquakes along identified tectonic structures (Jouanne et al., 2001; Walpersdorf et al., 2018; Nocquet et al., 2016). Paradoxically, leveling measurements indicate uplift rates up to 2.5 mm/year in the northwestern Alps (Nocquet et al., 2016). Focal plane mechanisms show that the inner Alps are characterized by extensional stresses, whereas the external Alps, including the studied area, are still under compression (Sue et al., 1999; Delacou et al., 2005). Within these compressional areas, plateau surfaces at 150-400 m above the present rivers suggest active uplift of the Western Alps due in part to flexural isostatic response to Quaternary erosion (Champagnac et al., 2007, 2008) and in part to tectonic processes (Collina-Girard and Griboulard, 1990; Schwartz et al., 2017).

The flexural history of the Alps is particularly recorded in the Neogene basins at the front of the Western Alps (e.g. Beck et al., 1998), such as the Digne-Valensole Basin in the southern Alps (Mercier, 1979). The Digne-Valensole Basin collected material eroded from the Alps and transported by the Durance, Bléone, Asse and Verdon rivers (Figure 2). The basin is filled by marine deposits overlain by conglomerates of the continental Valensole Formation, interpreted as an alluvial fan system prograding southward (e.g. Clauzon et al., 1989). Deposition starts with Aquitanian marine sandstone, followed by the continental conglomerate of the Valensole Formation which is Serravallian to Tortonian in age at its base and up to the early Pleistocene at its top (Mercier, 1979; Clauzon et al., 1989; Dubar, 1984a; Dubar et al., 1998). The eastern edge of the basin is overthrusted by the Middle Miocene to Late Quaternary Digne Nappe (Lickorish and Ford, 1998; Roure et al., 1992; Gidon and Pairis, 1988; Hippolyte et al., 2011), and it is bordered to the West by the NNE-trending Durance seismically active fault, a dextral fault with a reverse west-side-up component (Roure et al., 1992; Cushing et al., 2008). The Digne-Valensole basin is of particular interest for the study of the interaction between tectonics and surface processes because of its structural location and its Late Pliocene to Quaternary infill that allows to demonstrate the Quaternary activity of several faults within the basin, and along its borders (Clauzon, 1982; Ritz, 1992; Cushing et al., 2008; Hippolyte and Dumont, 2000; Hippolyte et al., 2011).

The Digne-Valensole Basin is dissected by the Bléone and the Asse rivers which divide the area in three parts : the Valensole

plateau to the South, the Puimichel plateau in the middle, which is partly eroded and incised by the Rancure River, and the mountains of the Duye valley to the north. These rivers also dissected the Valensole Basin during the Messinian crisis, when the Mediterranean sea-level dropped by about 1500 m (Ryan and Cita, 1978; Hsü et al., 1977; Clauzon, 1982; Clauzon et al., 2011). In the Valensole Basin, the Messinian paleo-canyons are mostly buried under Pliocene and early Quaternary sediments. A few sections of these paleocanyons could be mapped near Oraison (Dubar et al., 1998) and near Digne (Hippolyte et al., 2011), where the Messinian erosional surface separates the Valensole I and II Formations (Dubar, 1984a; Clauzon, 1996). The Valensole-II Formation filled the Messinian canyons and covered the central and southern part of the Valensole basin (Dubar, 1984a), its top surface presently forming the Valensole and Puimichel plateaus (Figure 3). The age of the top surface of these plateaus ranges between 0.7 Ma in the East, near the Digne Thrust (Dubar et al., 1998), and 1.7 Ma in the West, along the Durance River (Dubar, 1984a; Dubar and Semah, 1986; Dubar, 2014).

This surface was used by Champagnac et al. (2008) as a passively deformed marker to identify long-wavelength tilting of the Alpine foreland, in part as a response to erosional unloading. At shorter wavelength, the exceptional preservation of this surface allowed to demonstrate the Quaternary activity of the Lambruissier anticline (Figure 1A) within the Digne-Valensole Basin (Hippolyte and Dumont, 2000). This SW-vergent fold generated a 80 m high and 5-km long morphological ridge above the Puimichel plateau surface. Younger terraces have been mapped in the area (Dubar, 1984b), but due to active erosion and poor preservation of their surfaces it is not possible to use them as reliable benchmarks to measure finite deformation. To the North-East of the Lambruissier anticline, an older, Late Neogene fold is stratigraphically overlain by horizontal Late Pliocene deposits of the Bléone river messinian canyon (Hippolyte et al., 2011). These ages demonstrate the southward propagation of deformation within the Digne Valensole basin, which would be further confirmed if the Mées ramp anticline (Dubar et al., 1978), located South of the Lambruissier anticline, was active. This activity is suggested by (1) elevation anomalies in the Mindel terrace of the Durance river (Gabert, 1979; Dubar, 1984b), (2) the dip of the Puimichel Plateau surface which is more than three times higher (25 m/km) than the dip of the Valensole Plateau (8 m/km) (Hippolyte and Dumont, 2000) and (3) the striking asymmetric pattern of the drainage network that may have recorded a progressive tilt of the Puimichel Plateau (Collina-Girard and Griboulard, 1990; Hippolyte and Dumont, 2000). Recent tectonic deformation of this area is also in agreement with modelling of the thermal history of the northern tip of the Digne-Valensole basin (Schwartz et al., 2017).

Our study is specifically focused on the western edge of the Puimichel Plateau, which is dissected by a series of small basins (Figure 2) draining directly into the Durance river. The eastern limit of these basins corresponds to the post-Pliocene abandonment surface of the summit of the Plateau. Seismic surveys and drilling have identified an important uplifted basement structure below this plateau (Mées Structure, Figure 2C) of Upper Cretaceous to Eocene age, with possible Alpine Miocene reactivation (Dubois and Curnelle, 1978). The region is characterized by a Mediterranean climate with Mean Annual Temperature (MAT) of 13$^o$C and Mean Annual Precipitation of 700 mm (data at Saint Auban Météo France weather station over the 1981-2010 period). The dominant lithology is the Mio-Pliocene Valensole conglomerate, with an age either pre- or post-Messinian depending on the geometry of erosional surface, which has not been continuously mapped in this area. However, bedrock geology is uniform, with mostly clast-supported conglomerates, which weather primarily by destruction of the sandy matrix over a few tens of centimeters below the surface. The clasts (5 to 10 cm in size) are set loose but remain interlocked with little vertical

movement and mixing inside the regolith profile. Once the clasts reach the surface they are free to move downslope.

For the hillslope domain, the existence of this extensive mobile regolith cover implies that hillslopes are mostly under transport-limited regime, which is an important requirement of the topographic analysis described below (Figure 3). Concerning the fluvial network, we did not observed any large-scale alluvation but only a thin cover of sediment with many occurrences of outcropping un-weathered conglomerate bedrock, which leads us to consider that these streams are mostly under detachment-limited dynamics.

## 3   Methods

### 3.1   Topographic analysis

The topographic analysis carried out in this study relies on a 1 m resolution airborne LiDAR Digital Terrain Model (DTM) acquired in 2014 as part of the RGE ALTI® database from IGN (Institut National de l'information Géographique et Forestière) and covering our study area, as well as most of the Valensole and Puimichel Plateaus. The core of our analysis consists in the extraction of high-resolution hillslope and channels topographic metrics along a transect located at the northwestern edge of the Plateau in order to identify short-wavelength variations in the distribution of surface processes (Figure 2), as opposed to the long-wavelength deformation investigated by previous studies (Champagnac et al., 2008).

#### 3.1.1   Hillslope morphology

We present here the theoretical background supporting the interpretation of hillslope-scale morphological parameters. Mass conservation across a steady-state 1D hillslope profile can be expressed as,

$$\frac{\partial q_s}{\partial x} = \beta E, \tag{1}$$

where $x$ is the horizontal coordinate ($[L]$), $q_s$ is the sediment flux ($[L^2T^{-1}]$), $\beta$ is the rock-to-regolith density ratio (dimensionless), and $E$ is the erosion rate ($[LT^{-1}]$) which is equal to the rock uplift rate under steady state conditions. Equation 1 can be combined with a Geomorphic Transport Law (GTL, Dietrich et al., 2003), describing sediment flux $q_s$ over a hillslope as a non-linear function of local hillslope gradient $S = \partial z/\partial x$ (Roering et al., 1999, 2007),

$$q_s = \frac{DS}{1 - \left(\frac{S}{S_c}\right)^2}, \tag{2}$$

where $D$ is a diffusion coefficient ($[L^2T^{-1}]$) and $S_c$ a critical hillslope gradient (Roering et al., 1999). For gentle slope areas, such as the vicinity of hilltops, equation 2 can be linearly approximated as $q_s = DS$. Combining this expression of $q_s$ with equation 1 yields a linear relationship between the erosion rate $E$ and the second spatial derivative of topography or hilltop curvature $C_{HT}$,

$$E = \frac{DC_{HT}}{\beta}. \tag{3}$$

Equation 3 will be central in the interpretation of our results, as it allows to combine the two types of data acquired at hilltop sites during this study : high-resolution morphometric measurements ($C_{HT}$) and denudation rates ($E$) derived from cosmogenic nuclides concentrations.

Combining equations 1 and 2 and integrating yields the steady state elevation profile $z$ associated with a spatially uniform erosion rate $E$ (Roering et al., 2001),

$$z(x) = \frac{DS_c^2}{2\beta E} \left( \ln \left( \frac{1}{2} \sqrt{1 + \left( \frac{2\beta E x}{DS_c} \right)^2} + \frac{1}{2} \right) - \sqrt{1 + \left( \frac{2\beta E x}{DS_c} \right)^2} + 1 \right). \tag{4}$$

With $L_H$ as the hillslope length (horizontal distance from hilltop to channel), a reference erosion rate (Roering et al., 2007) can be defined as,

$$E_R = \frac{DS_c}{2L_H\beta}, \tag{5}$$

and similarly a reference relief $R_R = S_c L_H$ which represents the maximum hillslope relief.

These two reference values allow to normalize hillslope relief $R$ and erosion rate $E$ into their non-dimensional equivalent as,

$$R^* = \frac{R}{S_c L_H}, \tag{6}$$

and dividing equation 3 by equation 5,

$$E^* = \frac{2C_{HT}L_H}{S_c}. \tag{7}$$

Finally, equation 4 can be expressed in non-dimensional form,

$$R^* = \frac{1}{E^*} \left( \sqrt{1 + (E^*)^2} - \ln \left( \frac{1}{2} \left( 1 + \sqrt{1 + (E^*)^2} \right) \right) - 1 \right). \tag{8}$$

The purpose of our analysis of the DTM is to extract these various metrics characterizing the relief structure and erosion of hillslopes, and in particular the hilltop curvature $C_{HT}$, as well as hillslope relief $R$ and length $L_H$. These measurements will then allow to compute a non dimensional erosion rate $E^*$ according to equation 7. We use the approach presented by Hurst et al. (2012) and Grieve et al. (2016b), which we implemented into the GRASS GIS environment (Neteler et al., 2012) and R scripting language (R Core Team, 2018), as described in Godard et al. (2019) (Figure 4).

First, the DTM was filtered to remove short-wavelength noise using the despeckling algorithm of Sun et al. (2007) and its application to DTM filtering as described in Stevenson et al. (2010). The river network was extracted using a geometric approach by defining a 0.2 m$^{-1}$ contour curvature threshold for the definition of channel heads (e.g. Pelletier, 2013; Clubb et al., 2014). The narrow floodplains present in our working area were delineated following the approach proposed by Clubb et al. (2017). Hilltops were then identified as the intersecting margins of basins over all range of stream orders and curvature was computed at every hilltop pixel by fitting a quadratic surface over a 30-m-wide window, which is large enough to filter-out short-wavelength surface roughness and small enough to avoid perturbation from the ridge-and-valley topographic signal (Hurst et al., 2012; Godard et al., 2016; Grieve et al., 2016c; Lashermes et al., 2007). Flow was routed downslope from hilltop

pixels to the edge of floodplains using the algorithm of Mitasova et al. (1996), and the resulting flowlines were used to compute hillslope relief and length (Hurst et al., 2012; Grieve et al., 2016a, b). Flowlines and associated data were grouped into patches of a least 50 contiguous hilltop pixels (Grieve et al., 2016b).

### 3.1.2 Channel morphology

We also extracted standard metrics from the river profiles of the studied catchments (Figure 5). River incision $I$ ($[LT^{-1}]$) can be parameterized as a function of along-channel topographic gradient $S$ and drainage area $A$ ($[L^2]$) as,

$$I = KA^m S^n, \tag{9}$$

where $K$ is an erodibility coefficient ($[L^{1-2m}T^{-1}]$), and $m$ and $n$ are empirical exponents (Howard, 1994; Whipple and Tucker, 1999). Under steady state conditions river incision $I$ equals rock uplift $U$ and equation 9 can be reorganized as,

$$S = \left| \frac{dz}{dx} \right| = \left( \frac{U}{K} \right)^{\frac{1}{n}} A^{-\frac{m}{n}}. \tag{10}$$

$k_s = \left( \frac{U}{K} \right)^{\frac{1}{n}}$ and $\theta = \frac{m}{n}$ are referred to as the steepness index and channel concavity (Wobus et al., 2006; Kirby and Whipple, 2012) and are often determined by regression in a slope-area diagram. Steepness indexes are of particular interest in tectonic studies due to their direct dependence upon the rock uplift $U$, and under the assumption of constant erodibility $K$, they can be used to decipher relative spatial variation in $U$ (e.g. Kirby et al., 2003). In order to allow meaningful comparison between

channels of different concavities a reference $m/n$ value is chosen and used in the regression in order to obtain a normalized steepness index $k_{sn}$.

Calculating $S$ by differentiating the river long-profile often yield noisy data. Following Perron and Royden (2012), equation 10 can be integrated from baselevel, at position $x_b$, to $x$ as,

$$z(x) - z(x_b) = \int_{x_b}^{x} \left( \frac{U}{KA^m} \right)^{\frac{1}{n}} dx. \tag{11}$$

Under the assumption that $U$ and $K$ are spatially constant equation 11 becomes,

$$z(x) = z(x_b) + \left( \frac{U}{KA_0^m} \right)^{\frac{1}{n}} \chi, \tag{12}$$

where $A_0$ is a reference drainage area, and with,

$$\chi = \int_{x_b}^{x} \left( \frac{A_0}{A} \right)^{\frac{m}{n}} dx. \tag{13}$$

Equation 12 implies that in a $z$ vs $\chi$ diagram a steady state channel profile should plot a straight line, with a slope proportional

to steepness index.

For every investigated basin we searched the $m/n$ value (concavity) leading to the best linearization of the $\chi$-transformed river profile (Perron and Royden, 2012). We fixed the reference value according to the mean of observed $m/n$ value across all basins. We then again $\chi$-transformed the river profiles using this reference $m/n$ to compute normalized steepness indexes.

## 3.2 Cosmogenic nuclides

The topographic analysis methods described in the previous section allow us to identify relative spatial patterns for the intensity of surface processes such as surface denudation which can be used to infer rock uplift distribution. We used *in situ*-produced cosmogenic nuclide concentration measurements to constrain the absolute denudation rate values. Active fluvial sediments are present along the stream network of the studied catchments (Figure 2A), and could be sampled to derive basin-averaged denudation rates (von Blanckenburg, 2005). However, most of the surveyed channels displayed complex dynamics with localized occurrences of aggradation, splitting of the main channel and local colluvial inputs, such that the required hypothesis of a homogeneous contribution of the whole upstream area was most likely invalid. For that reason we rather sampled material directly at the hilltops, which allows a clear identification of the origin of the sampled material and a straightforward comparison with the topographic metrics extracted from the high-resolution DTM. As noted earlier, weathering principally affects the sandy matrix of the conglomerate, liberating the clasts which remain interlocked until they reach the surface, such that vertical mixing within the regolith is minimal at the hilltop.

Samples for $^{10}$Be and $^{26}$Al concentrations measurements were collected at 10 hilltop sites, by amalgamating 30 to 40 individual sandstone clasts derived from the bedrock conglomerate (Figure 2A). Samples were crushed and sieved to extract the 250-1000 $\mu$m fraction, which was submitted to sequential magnetic separation. The remaining fraction was leached with 37% HCl to remove carbonate fragments. The samples were then repetitively leached with $H_2SiF_6$ and submitted to vigorous mechanical shaking until pure quartz was obtained. Decontamination from atmospheric $^{10}$Be was achieved by a series of three successive leachings in concentrated HF, each removing 10% of the remaining sample mass (Brown et al., 1991). After addition of an in-house $^{9}$Be carrier, the samples were digested in concentrated HF. Be and Al were isolated for measurements using ion-exchange chromatography. $^{10}$Be/$^{9}$Be and $^{26}$Al/$^{27}$Al measurements were performed at the French AMS National Facility, located at CEREGE in Aix-en-Provence. All results and technical characteristics of the measurements and calculations are provided in tables 1 and 2.

## 3.3 Surface deformation modelling

Under the assumption that the spatial distribution in erosion rates results from tectonic forcing, observations of absolute or relative variations in the intensity of surface processes are commonly used to derive information on rock uplift patterns and the geometry of associated structures (Wobus et al., 2006; Scherler et al., 2014; Le Roux-Mallouf et al., 2015). While the key hypothesis of a tectonic origin for the variability in erosion proxies will need to be discussed in details, we present here a modelling approach which can be used to interpret surface erosion patterns, considered as proxies for rock uplift, in terms of the associated structures at depth.

We use a simple elastic dislocation model (Okada, 1985) to predict surface deformation distributions and compare them with our observed $E^*$ evolution along the investigated profile (Figure 2). This type of modelling approach is usually associated with the study of upper crustal deformation at the scale of the seismic cycle, but has also been successfully applied to interpret surface deformation patterns associated with blind thrusts over longer timescales (Ward and Valensise, 1994; Benedetti et al.,

2000; Myers et al., 2003). As noted by Myers et al. (2003), who used elastic dislocation modelling to study folding in the Los Angeles Basin, it allows to keep track at first order of the displacement of material associated with the activation of the fault, independently of the mechanical parameters. We consider a single planar dislocation embedded in an elastic medium.

We define its geometry with 4 parameters : the horizontal position and depth of its upper limit (varying from 0 to 20 km along the profile and from 0 to 10 below the surface, respectively), dip angle (varying from 0 to 90$^o$) and length (varying from 0 to 4 km). This set of 4 parameters allows to predict a surface deformation profile which is compared to the observed $E^*$ pattern. We do not fit the absolute value of $E^*$ which has no signification with respect to the elastic dislocation model, but rather the wavelength of the predicted deformation and simply scale its amplitude to that of the observed $E^*$ profile.

We use a standard Monte Carlo-Markov Chain (MCMC) approach to move through this parameter space and estimate the posterior distributions (Metropolis et al., 1953). The consecutive displacements during the sampling procedure were driven by the Metropolis-Hasting algorithm, with an acceptance rate of 20%. We ran 16 independent chains, each $10^6$, in length with a $10^4$ length burn-in phase. The multivariate Potential Scale Reduction Factor is 1.004 suggesting convergence was achieved (Gelman and Rubin, 1992).

## 4   Data

In this section we present the primary data acquired during this study and the associated direct observations. The interpretations built on these datasets are developed and discussed in the next section.

### 4.1   Topographic analysis

Most hillslope profiles display a clear convexity and progressive downward increase in slope, with almost linear portions at
295 the bottom of the hillslope and gradients close to 0.6 m/m (Figure 6A). Calculation of average hillslope gradient shows that most hillslope have average gradients below 0.6 m/m (Figure 6B). Following (Hurst et al., 2019), we determined the value of $S_c$ for which 99% of hillslopes have a relief inferior to the maximum value ($L_H S_c$), and obtained $S_c = 0.52$ m/m. In the following we use $S_c = 0.6$ m/m for the critical gradient value (Figure 6C). This $S_c$ value is lower than what was found in other settings (Grieve et al., 2016b), but is close to the natural angle of repose in many granular materials ($\sim 30^o$). We note that
two particularities of our study area are the nature of the regolith, which is mostly constituted of highly mobile conglomerate-derived clasts moving downslope by both creep and dry ravel, and the isolated vegetation providing little cohesion (Figure 3B), as opposed to the finer-grained soils supporting denser root networks observed in many other similar studies. Most topographic metrics extracted at the hillslope scale display important variations along the studied profile, both in terms of basin-averaged or binned values. Non-dimensional erosion rate $E^*$ increases 2-fold from South to North (Figure 7A). This variation is not
evenly distributed along the profile but occurs over less than 4 km of horizontal distance. The hilltop curvature $C_{HT}$ pattern closely mimics that of $E^*$, increasing from 0.01 to 0.02 m$^{-1}$ (Figure 7B). For both metrics, basin-averaged values are highly consistent from one catchment to the other and delineate a clear trend along the section, with the exception of one outlying small catchment at $\sim 4$ km distance. The evolution of hillslope relief is less pronounced, but also show an increase from $\sim 40$

to ∼60 m (Figure 7C). On the contrary no clear systematic changes in hillslope length can be observed along the profile, with values ranging from 140 to 160 m (Figure 7C).

For all studied basins the river profiles display a regular concave-up shape, and in most situation $\chi$-transformed main trunk profiles, as well as the main tributaries, collapse along a linear trend, suggesting the absence of major transient perturbation propagating through the river network (Figure 5). Small tributaries usually show higher dispersion due to changes in processes in small colluvial valleys. Usual topographic indexes, such as $m/n$ ratio and normalized steepness index ($k_{sn}$), were extracted from the fluvial network. $m/n$ ratio ranges from 0.2 to 0.4, with an average of 0.24±0.06 and a reference value of 0.25 was used in the following analysis. While this $m/n$ value is lower than the often reported 0.4 to 0.5 ratio, it is within the range of observations from Harel et al. (2016) for high erodibility lithologies, such as the setting we consider here. These fluvial metrics display a larger amount of scatter from one basin to another, when compared with the patterns extracted from the hillslope morphology analysis (Figure 7D). However, it can be noted that the 4 northernmost basins display higher $k_{sn}$ values than the rest of the section, and that basin-averaged $E^*$ and $k_{sn}$ are significantly positively correlated (Figure 8).

## 4.2 Cosmogenic nuclide data

Measured concentrations in our hilltop samples range from 43 to $117 \times 10^3$ at/g and 281 to $867 \times 10^3$ at/g for $^{10}$Be and $^{26}$Al, respectively (tables 1 and 2). The corresponding $^{26}$Al/$^{10}$Be ratios vary from 6.5±0.8 to 8.2±1.2. While some of these ratios are slightly higher than the theoretical value, $1\sigma$ uncertainties ellipses are always overlapping the steady state denudation curve (Figure 9). These concentrations correspond to denudation rates ranging from 42 to 115 mm/ka and from 38 to 114 mm/ka for $^{10}$Be and $^{26}$Al, respectively (tables 1 and 2). The denudation rates calculated from both nuclides concentrations are consistent but display a small deviation from the 1:1 line, with $^{26}$Al denudation rates slightly lower than their $^{10}$Be equivalent. The observed $^{26}$Al/$^{10}$Be values argue against a significant contribution of an inherited CRN inventory from the history of the clasts prior to their deposition inside the Valensole conglomerate, in particular if they derived from the Valensole II formation. In the following we only consider $^{10}$Be data for our analysis of the relationship between high-resolution topography and denudation rates, due to their lower uncertainty.

## 5 Discussion

We now discuss the morphological and geochronological data presented in the previous section, in terms of the main controls on hillslope morphology, the relationship between hillslope and channel properties, the geometry of the underlying tectonic structures, and the comparison between morphological observations and denudation rates at the same sites.

## 5.1 Interpretation of the spatial variability in hillslope morphology

### 5.1.1 Possible controls on hillslopes morphology

We observe a pronounced and systematic variation of hillslope morphology along the studied transect, with hillslope curvature $C_{HT}$ undergoing a 2-fold increase from S to N (between 7 and 10 km on Figure 7B). We evaluate below the possible controls on this evolution. We first note that our study area extends over ∼10 km in length, with catchment average elevation and relief ranging from 460 to 620 m and from 100 to 300 m, respectively. These limited changes in elevation imply that climatic conditions can be considered as constant in terms of Mean Annual Precipitation and Temperature (MAP and MAT), and cannot account for the observed variations in hillslope morphology. Vegetation cover is also homogeneous over the western flank of the Puimichel Plateau, with a forest dominated by *Quercus pubescens* and occurrences of *Quercus ilex* and *Pinus sylvestris*. Similarly, the investigated basins are eroding into Mio-Pliocene conglomerates with no major changes in the nature and properties of the bedrock or regolith material. We note that this homogeneity of geological, climatic and biological properties over the transect is a specificity of our studied area, and might not be warranted in other settings, where eventual disparities in these properties might complicate the interpretation of fluvial and hillslopes morphologies.

Hillslopes have been shown to record transient waves of erosion propagating through landscapes (Hurst et al., 2012, 2013a; Mudd, 2017). The series of studied basins are directly connected to the Durance river baselevel, and the eventual propagation of incision pulses and knickpoints alongstream might impact the network of tributaries and adjacent hillslopes, inducing a differential hillslope response (Hurst et al., 2012). However, several lines of evidence argue against such control on the observed distribution of hillslope properties. First, no major knickpoints have been identified along the Durance river in the vicinity of our working area. Second, we note that the pattern of evolution for $E^*$, $C_{HT}$ and $R$ along the transect is characterized by almost constant values for 6 km, followed by a rapid increase over less than 4 km, rather than the progressive variations that would be expected to result from the propagation of a knickpoint in front of the western edge of the Plateau. At last, we note that the propagation of an incision wave would result in a pattern with higher erosion areas in the southern part connected to the adjusted or adjusting landscape downstream of the propagating incision pulse, and slower erosion in the northern yet unaffected areas, which is exactly contrary to what we observe here. Therefore, we propose that the observed pattern is unlikely to result from the transient adjustment of the landscape to the propagation of a wave of incision along the Durance river.

Another possibility to generate the observed distribution of hillslope parameters would be a sustained differential rock-uplift pattern associated with recent or ongoing deformation. Several studies have already pointed at geomorphic evidence for recent tectonic activity in the northern part of the Puimichel Plateau (Collina-Girard and Griboulard, 1990; Hippolyte and Dumont, 2000), which is confirmed by recent seismic activity (Nicolas et al., 1990), with several magnitude 4 thrust-slip events with EW strike directions (Figure 1B). Notably, a major inflection of the abandonment surface occurs at 7-8 km of horizontal distance along the transect (Figure 2B), suggesting post-Pliocene uplift of the northern part. The location of this surface inflection coincides with the transition area for $E^*$, $C_{HT}$ and $R$ (Figure 7). Such spatial coherence between a long-term finite deformation pattern passively recorded at 1 Ma time-scale by the summit surface, and the shorter-term active erosional response of the landscape observed through hillslope morphometric indices, argue for a control by differential uplift rates across the transect.

Additionally, we note that this transition zone is also coincident with the southern flank of a major basement uplift, identified by seismic surveys, located below the Northern part of the Puimichel Plateau (Figure 2C and 2D). This uplifted basement is a long lasting structure formed during the Pyrenean orogeny and is associated with basement thrusts on its southern edge, at the location of the observed geomorphic transition (Dubois and Curnelle, 1978). Quaternary reactivation of such faults has been invoked to explain the surface deformation pattern of the Plateau farther to the North (Hippolyte and Dumont, 2000), and we

propose a dominant tectonic control for the distribution of proxies for surface denudation along our transect (Figure 7).

The $R^*$ vs $E^*$ relationship (with $S_c = 0.6$ m/m) shows that the studied catchments plot slightly below the steady state line predicted by equation 8 (Figure 10), indicating possible decaying dynamics of the landscape toward lower relief, likely due to a change in the climatic or tectonic boundary condition (Mudd, 2017). All catchments appear to be similarly affected, with no obvious gradient between the northern and southern ones. For that reason, this decay is not likely to result from a decrease in

the amount of differential rock uplift across the transect, but could be associated with a regional change in the intensity of the top-down forcing, of climatic origin, which would modify the value the diffusion coefficient $D$. However, we note that using $S_c = 0.5$ m/m, which appears to be reasonable for the vast majority of the studied hillslopes (Figure 6), reduces considerably the deviation from the steady state line.

### 5.1.2 Constraints on tectonic structures from hillslope morphology

As discussed in the previous section, the hypothesis that the evolution of hillslope morphology along the studied transect (Figure 7) results from a spatial variations of rock uplift is supported by several lines of evidence and notably the presence of a coincident major warping of the Pliocene abandonment surface of the plateau. Here we develop further the interpretation of this surface deformation pattern, in order to put constraints on the geometry of the associated structures. We use the dislocation modelling approach presented above to constrain the geometry of a fault whose displacement could explain the observed

change in $E^*$ along the transect (Figure 7) through an inversion procedure. The parameters characterizing this fault geometry are the horizontal position and depth of the upper tip of the fault, its dip angle and length (Figure 11).

We observe that the horizontal position of the top of the fault is the best constrained parameter with a most likely value around 8 km (same horizontal reference frame as the profile on figure 7). The most likely depth for the upper limit of the dislocation is around 2 km and and the suggested dip of the structure is $>50^o$. The length of the dislocation is not well constrained by our

inversion.

Interestingly, the suggested horizontal position for the upper tip of the dislocation is close to the major deflection of the Pliocene abandonment surface, imaged by the LiDAR data (Figure 11B). The high-$E^*$ northern part of the studied transect roughly corresponds to the Mées structure identified from seismic surveys and drilling (Dubois and Curnelle, 1978), which is a large anticline inherited from the late-Cretaceous-Eocene compression of the Pyrenean phase (Figure 2B). We note that the

400 suggested horizontal position of the dislocation is also coincident with the southern flank of the structure and the complex of thrusts responsible for the folding. Some of these thrusts correspond to reactivated high-angle basement structures, which is in agreement with the inferred dip angle of the dislocation. The depth of at least 2 km for the top of the dislocation is also consistent with the observation that the Mio-Pliocene reflectors do not display major offsets in the available seismic data (Dubois and

Curnelle, 1978), and that the Cenozoic formations underwent long-wavelength folding rather than localized faulting. Overall, the geometry of the structure constrained by our simple model is compatible with the activity of steep south-verging inherited structures affecting the basement and Mesozoic series, and the Quaternary reactivation of which induced a long-wavelength warping of the Mio-Pliocene cover and differential uplift along our transect. We note that several other recently active structures have been documented farther to the North, corresponding to similarly oriented south-verging thrust-and-fold systems, such as the Lambrussier anticline which affects the northern edge of the Puimichel Plateau (Hippolyte and Dumont, 2000). The amount of finite deformation accommodated by these folds progressively decreases southward, such that the structure we identified could correspond to the most recently activated as an in-sequence system. Finally, while high-resolution hillslope morphology analysis has already been used to constrain rock uplift patterns in a limited number of studies (Hurst et al., 2013a, 2019; Clubb et al., 2020), our results are the first to illustrate the use of such data to infer the geometry of tectonic structures at depth.

## 5.2  Hillslopes and channel dynamics

Interestingly, parameters extracted from the analysis of river long-profiles such as steepness indexes, which are commonly used to decipher tectonic patterns in erosional landscapes (Kirby and Whipple, 2012), do not display as clear a pattern as hillslope metrics (Figure 7D). Normalized steepness index values ($k_{sn}$) are in average higher in the northern part of the transect with respect to the southern part and positively correlated with $E^*$ (Figure 8), but the data is scattered and we do not observe a clear progressive increase comparable to what is displayed by $E^*$ or $C_{HT}$. In each of the studied basins, the main trunk and its tributaries display regular concave up profiles, which collapse along a single trend in $\chi$-plots (Figure 5). This observation suggests that, at the scale of each catchment, the river network is globally equilibrated with respect to a common rock uplift rate (Perron and Royden, 2012), and argue against the impact of local perturbation along the river profile as an origin for the observed scatter. An underlying assumption of our river profile analysis is that the streams behave as purely detachment-limited systems. While our field survey did not allow us to identify thick alluvial cover along the stream network, local and intermittent shifts toward transport-limited behavior could explain the apparent subdued response of $k_{sn}$ across the inferred rock uplift gradient. We note however that in such situation, stream concavity ($m/n$) should display some sensitivity to changes in rock uplift (Wickert and Schildgen, 2019), which is not observed here (Figure 7D).

A key difference between the two approaches is the very high spatial density of the metrics extracted for the hillslope dataset, which is several orders of magnitude denser than the evaluation of steepness index and concavity at 18 basins. This comparison illustrates the resolving power of high-resolution hillslope morphology analysis, which allows to document short-wavelength patterns of erosion and uplift that are undersampled by scarcely distributed fluvial metrics. We note that Hurst et al. (2019) observed a clear response of both hillslope and channel metrics across the tectonic gradient they studied in the vicinity of the San Andreas Fault. Two important differences with our study are the existence of transient channel adjustment along the Bolinas Ridge and the dimension of the section, which is $\sim$30 km long in their case, compared to the $\sim$10 km of our profile. Combined with a longer wavelength of the underlying tectonic signal, this latter difference might be a reason for the better sampling through fluvial metrics by Hurst et al. (2019), and the clearer relationship they observe with hillslopes properties.

One prominent feature of the hillslope evolution across the rock-uplift gradient is the lack of significant changes in hillslope length, contrasting with the other extracted metrics (Figure 7). Hillslope length $L_H$ remains nearly constant across the transect between 140 and 160 m, whereas hilltop curvature $C_{HT}$, which can be considered as a proxy for erosion and rock uplift under a steady-state assumption, undergoes a nearly two-fold increase. There is no significant inverse correlation between $L_H$ and $C_{HT}$ as observed by Hurst et al. (2013b). The characteristic horizontal lengthscale of landscapes, which can be evaluated through different types of measurements, such as drainage density, spacing of $1^{st}$ order valleys or hillslope length, has been shown to be highly sensitive to external tectonic and climatic forcings, as well as internal parameters controlling erosion processes (e.g. Perron et al., 2008a; Pelletier et al., 2016; Clubb et al., 2016; Hurst et al., 2019). For example, Clubb et al. (2016) studied in detail the relationship between denudation rate and drainage density with analytical and numerical models as well as high-resolution topographic and cosmogenic nuclide data. They show a sensitivity of drainage density to erosion rates, which is very pronounced in the 50-100 mm/ka range corresponding to our CRN data (Figure 9), and conflicts with the observation of a nearly constant $L_H$ along our transect. However, this relationship is highly dependent on the parameters used for the fluvial and hillslope erosion laws, as shown by the formulation of the landscape Péclet number proposed by Perron et al. (2008b, 2009) as the ratio between characteristics fluvial and hillslope timescales :

$$P_e = \frac{K l^{2(m+1)-n}}{D \zeta^{1-n}}, \tag{14}$$

where $K$ is the erodibility parameter for fluvial incision, $D$ the hillslope diffusion coefficient, $m$ and $n$ the area and slope exponents of the fluvial incision law respectively, and $l$ and $\zeta$ horizontal and vertical lengthscales for the landscape. In the special case where the slope exponent $n$ of the stream power formulation for river incision is equal to 1, this Péclet number becomes independent from relief $\zeta$, and hence uplift rate. Considering $P_e = 1$ allows to retrieve $l_c$ as a characteristic lengthscale for hillslope/channel transition, which again, in the $n = 1$ case, does not depend on erosion or uplift rates. Therefore, the stability of $L_H$ across a two-fold erosion rate gradient, observed in our dataset, would hint toward a value of $n$ close to unity. While the ratio $m/n$ can usually be constrained from the measured concavity of river profiles, the absolute values of the slope and area exponents of the stream power description for fluvial incision are debated, with many evidences pointing to $n > 1$ (DiBiase and Whipple, 2011; Lague, 2014; Harel et al., 2016). However, it is noteworthy that values closer to unity have often been reported for high erodibility sediments (Harel et al., 2016) or small catchments affected by colluvial processes (Lague and Davy, 2003), which are both notable characteristics of the area we investigate. In any case, our study clearly illustrates the potential of high-resolution hillslope morphological properties to resolve short-wavelength variations in rock uplift that are difficult to capture from the conventional methods based on fluvial profiles analysis.

## 5.3 Comparison of cosmogenic nuclides data with high-resolution topography metrics

Our understanding of landscape dynamics relies on the formulation of Geomorphic Transport Laws (GTL), such as equation 2, as the foundation of Landscape Evolution Models (Dietrich et al., 2003). In most situations such models can be expressed as a differential equation involving spatial and temporal derivatives of the topographic surface elevation. More precisely, these equations will often relate the spatial structure of the landscape involving topographic slope or curvature with its rate of evolu-

tion as, for example, hillslope denudation rate or fluvial incision. Evaluating the relevance of such models requires obtaining actual measurements for these spatial and temporal descriptions of the landscape. Spatial properties of landscapes are usually derived from the analysis of DEM at various resolutions, from which topographic gradient and curvature can be computed. On the other hand, geochronology techniques, such as cosmogenic nuclide concentration measurements in bedrock or sediment

samples, allow constraining the rate of lowering of the topographic surface though time, and provide the framework to evaluate the temporal component of the landscape evolution problem.

For example, the comparison of Catchment-Wide Denudation Rates (CWDR) calculated from measured [10]Be concentrations in river sediments with topographic gradient extracted from Digital Elevation Models has provided critical tests of GTL for hillslope sediment flux (Ouimet et al., 2009). While not often explicitly formulated in terms of the evaluation of a GTL, this

kind of connection between some spatial properties of landscapes (slope, relief, steepness index, etc ..) and their rates of evolution is now standard in CWDR studies. However, it is to be noted that the interpretation of CRN concentrations in river sediments in terms of spatially averaged denudation rates suffer from important limitations resulting from the heterogeneity and stochasticity of erosion processes in space and time (e.g. Yanites et al., 2009). Furthermore, the overwhelming majority of these studies rely on the evaluation of topographic metrics derived from medium resolution DEMs (pixel size $>10$ m) for

which the computation of the spatial derivatives controlling erosion rates and sediment fluxes is inaccurate at the scale of hillslopes. Only a handful of studies have actually attempted to reconcile CRN-based denudation data with topographic metrics extracted from High Resolution DTMs into a GTL-based physical framework (e.g. DiBiase et al., 2012; Hurst et al., 2012, 2013b; Godard et al., 2016, 2019; Neely et al., 2019).

Our dataset allows carrying out such comparison of CRN denudation rates determined at hillslope sites with high-resolution

hillslope topographic properties. In particular, we test the consistency of our dataset with prediction of simple hillslope diffusion formulations such as equation 3 relating hilltop curvature $C_{HT}$ with denudation rate. We observe that no single diffusion coefficient can explain the distribution of our data, but that at most sites, the values of $C_{HT}$ and erosion rates are compatible with $D$ in the 0.003 to 0.006 m$^2$/a range (Figure 12). The very high denudation rate observed for sample P, and resultant high diffusion coefficient, can be a consequence of a recent anthropogenic disturbance, with the presence of small walls made from

collected cobbles and possible shallow excavation in the vicinity of the sampling site. This data point is not further considered in the following analysis. The range of observed diffusion coefficients is consistent with values reported by available compilations for this similar lithologies and climate (Mean Annual precipitation of 700 mm) (Hurst et al., 2013b; Richardson et al., 2019). There is no climatic gradient over the limited extent of our study area, so this can not be invoked as possible control for the $\sim 2$ fold variations for the estimates of $D$ in our dataset, which in any case does not present a clear spatial pattern or clus-

tering. Similarly, bedrock geology is homogenenous between the different sites, with conglomerates of Miocene and Pliocene ages releasing clasts of an homogeneous 5-10 cm size group. We amalgamated up to 40 clasts at each sampling sites and carefully selected well-defined and near horizontal ridges with negligible topographic shielding, such that we do not consider that this spread in $D$ can arise from a systematic sampling bias. We note that the amplitude of variability for our estimates of $D$ at individual hilltops is similar to the uncertainty range for $D$ from studies based on CWDR (e.g. Hurst et al., 2013b). We

consider the distribution of the observed $D$ values to be mostly controlled by the natural variability of the hillslope processes

and persistent internal transience at the $\sim$100 m wavelength. Such variability, observed here at individual hilltop locations, is usually averaged out in studies relying on CWDR rather than local estimations based on bedrock CRN inventories.

We use the estimated distribution for $D$ to convert the $C_{HT}$ pattern along the transect into denudation rates using equation 3. We repeatedly sample the cumulative distribution for $D$ and use $C_{HT}$ aggregated for hillslope patches. The resulting values for denudation rates are binned at 1 km interval along the transect (Figure 12). The interquartile ranges of the bins are largely overlapping, but a systematic increase can still be observed, consistent with the underlying variation in $C_{HT}$. The estimated median denudation values are $\sim$30 and $\sim$50 mm/ka in the southern and northern part, respectively, which is comparable to denudation rates reported on surrounding landscapes (Siame et al., 2004; Thomas et al., 2017, 2018). We note that the denudation rate values measured at the various sites (blue circles on figure 12C) display an overall similar increasing trend, but with systematically higher values. This deviation results from a sampling bias toward high curvature ridges due to better field conditions at such sites. Indeed, measured $C_{HT}$ at these sites are $\geqslant 0.02 \, \mathrm{m}^{-1}$ (Figure 12A), whereas continuously averaged $C_{HT}$ along the transect are in average below $0.02 \, \mathrm{m}^{-1}$ (Figure 7B). Transport-limited conditions were prevailing at all surveyed sites such that we have no reason to question the validity of equation 3. For that reason the predominance of high-curvature/high-denudation sites in the dataset should not introduce a systematic deviation in the calibration of the diffusion coefficient $D$.

Under a steady state assumption we can propose that the observed 21±17 mm/ka differential denudation rate between the southern (distance < 5 km on Figure 12C) and northern (distance > 10 km) parts of the transect reflects a similar differential rock uplift rate across the transect. Considering a dip angle of 64±11$^o$ (Figure 11E) for the hypothetical fault responsible for the uplift pattern, it converts into a slip rate of 23±20 mm/ka, which implies that slip rate is most likely <50 mm/ka on this blind thrust. This slow slip rate estimation is consistent with observations on slowly slipping faults of Western Provence, where deformation usually cannot be resolved from geodetic data, and proposed long-term slip rates are < 100 mm/ka. For example, on the Trevaresse ridge fault, which produced the last major earthquake in metropolitan France (1909) with an estimated $M_w = 6$ (Baroux et al., 2003), trenches have yielded a Late Quaternary slip rate in the 50 to 300 mm/ka range (Chardon et al., 2005), whereas accumulated deformation since the late Miocene indicates a slip rate of 30±20 mm/ka (Chardon and Bellier, 2003). For the Middle Durance Fault, which is located directly West of our study area, across the Durance river, a slip rate ranging from 10 to 70 mm/ka has been proposed over the last Ma (Siame et al., 2004). At last, the uplift rate we calculated is of the same order of magnitude as the one reported for the Lambrussier anticline by Hippolyte and Dumont (2000), directly North of our study area. In these slow tectonics landscapes erosion processes often outpace uplift rates, eroding passive deformation markers. While some aspects of the investigated area, such as the lithology, size and orientation of our basins, present favorable characteristics, our study illustrates the potential of tracking active erosion processes through hillslope morphology analysis to retrieve tectonic information in this type of environments.

## 6  Conclusions

Our analysis of hillslope properties along a transect into the Digne-Valensole basin, at the Southern Alpine front, allows to identify an important systematic variation across a short horizontal distance (<10 km). In the absence of any climatic, litho-

logic or vegetation gradient, the observed increase in hilltop curvature, hillslope relief and normalized erosion rate points to a

540 coincident increase in rock uplift. Hillslope lengths appear to be constant along the studied transect and thus unaffected by this major change in the tectonic boundary condition. Usual metrics associated with channel profile geometry, such as normalized steepness index, only capture the first order change along the profile. Our results illustrate the utility of high-resolution analysis of hillslope morphology in low uplift areas. Such techniques have the advantage to provide a dense spatial coverage, whereas approaches based on the fluvial metrics are inherently limited by the geometry and distribution of the river network.

We also demonstrate that, using simple deformation models, the relative uplift pattern resolved from high-resolution hillslope morphology analysis can be used to constrain the geometry and activity of underlying tectonic structures. We merged the morphological information obtained for hillslopes morphology, with estimation of denudation rates from cosmogenic nuclides concentration measurements, to evaluate the diffusion coefficient for hillslope sediment transport. In addition to the investigation of relative rock uplift patterns allowed by the approaches described above, the combined use of cosmognenic nuclides

derived denudation rates at selected hilltop sites can be used to evaluate the amount of differential rock uplift across the transect. Such direct comparison of the spatial (high-resolution morphological analysis) and temporal (cosmogenic nuclides) aspects of landscape evolution has only be explicitly addressed by a limited number of studies and holds tremendous potential to decipher the response of various landscape elements to tectonic and climatic forcings.

*Author contributions.* VG carried out the topographic analysis, processed the CRN samples and developed codes used in the interpretation.

VG and JCH conducted field work. AT performed the AMS measurements. JCH, EC, NE, JF, OB and VO contributed to the understanding of the tectonics and geomorphology of the study area. VG prepared the manuscript with contributions from all co-authors.

*Competing interests.* The authors declare that they have no conflict of interest.

*ASTER Team is Georges Aumaître, Didier L. Bourlès, Karim Keddadouche

*Acknowledgements.* This study was supported by the ECCOREV research federation. This work is also a contribution to the Labex OT-

560 Med (ANR-11-LABX-0061) funded by the French Government "Investissements d'Avenir" programme of the French National Research Agency (ANR) through the A*MIDEX project (ANR-11-IDEX-0001-02). The $^{10}$Be and $^{26}$Al measurements were performed at the ASTER AMS national facility (CEREGE, Aix en Provence) which is supported by the INSU/CNRS, the ANR through the "Projets thématiques d'excellence" program for the "Equipements d'excellence" ASTER-CEREGE action and IRD. Some of the computing for this project was performed on the OSU Pythéas HPC cluster. Insightfull reviews by Martin D. Hurst, Marta Della Seta and Peter van der Beek allowed to

565 greatly improve our manuscript.

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

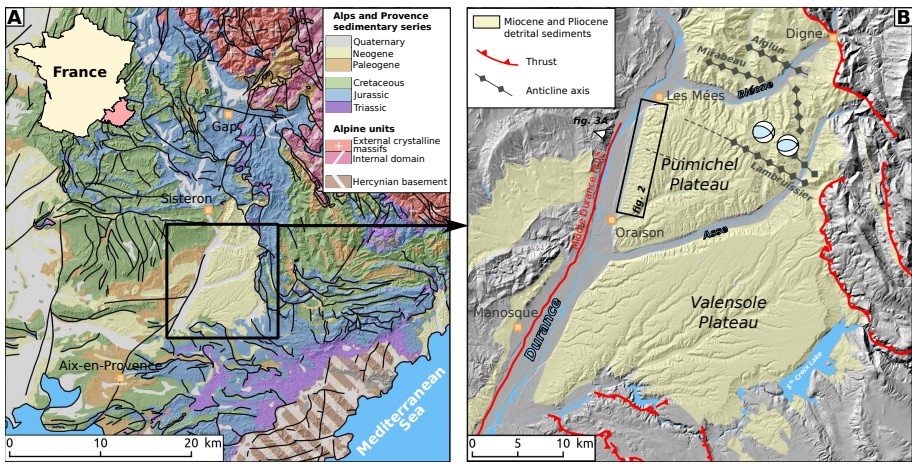

**Figure 1.** (A) Geological setting of the studied region in SE France (1/10$^6$ BRGM Geological map). Black square indicates the position of inset B. (B) Focus on the Mio-Pliocene Valensole Plateau and the main regional tectonic structures. Focal mechanisms from Nicolas et al. (1990).

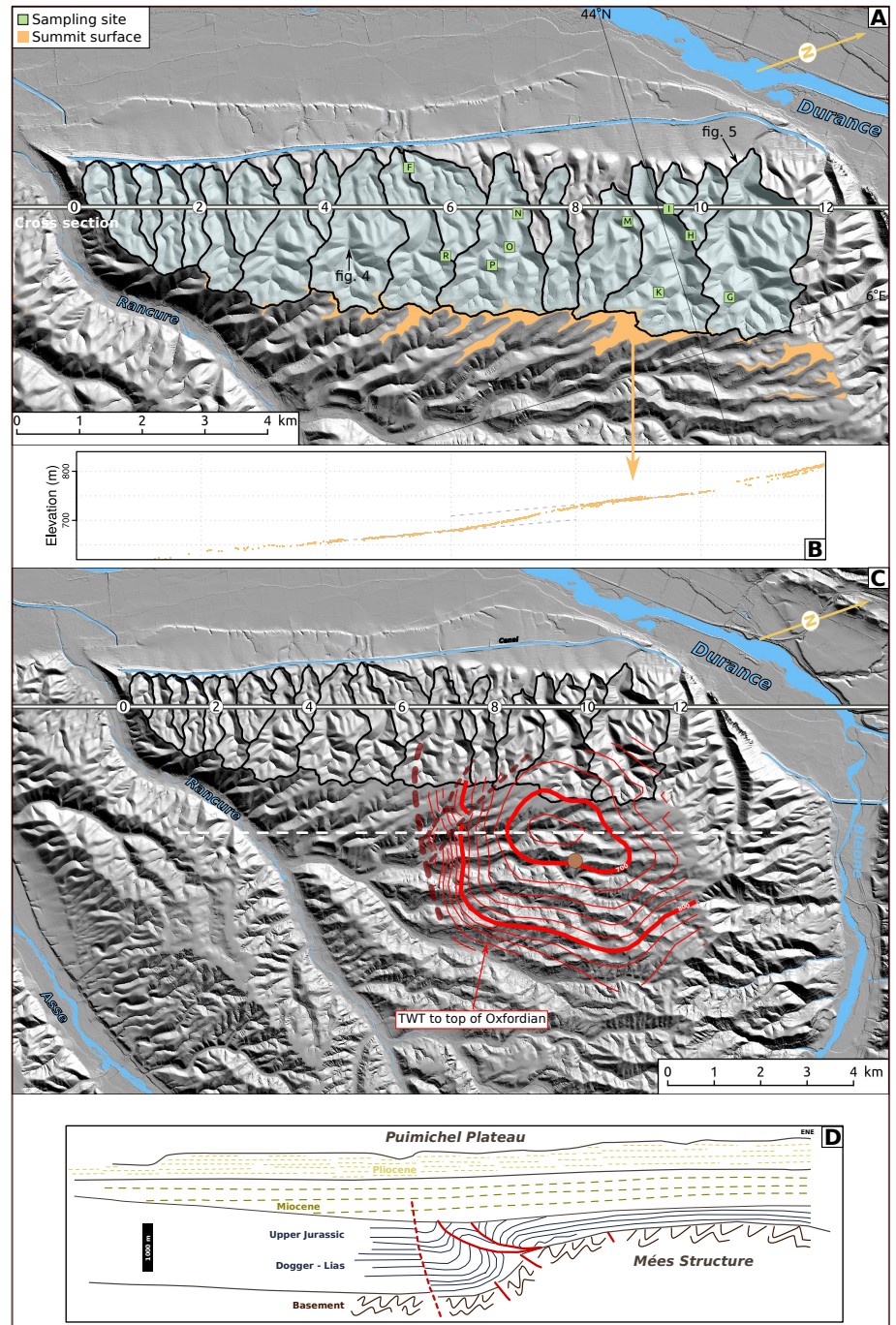

**Figure 2.** (A) Location of the studied basins and sampling sites at the western edge of the Puimichel Plateau (Figure 1). White line is the location of the profile used in figure 7. The numbers denote distance along the profile in kilometers and match the horizontal distance coordinate of figures 7 and 11. (B) Projection of the summit surface along the profile. Horizontal coordinates are consistent with labels of the cross section in panel A. (C) Red contour lines of the two ways travel times (ms) to the top of Oxfordian black marls from a seismic surveys synthesis included in the scientific report of drilling BSS002DWDJ available in the subsurface BSS database of BRGM. Brown circle indicates the location of the drill site. Dark red dashed lines delineate possible geological structures. Dashed white line is parallel to the main cross section and used as an additional section line in figure 11. (D) Simplified geological section across the Puimichel Plateau adapted from Dubois and Curnelle (1978). Note that the orientation of this cross section is slightly different from the geomorphological transect studied

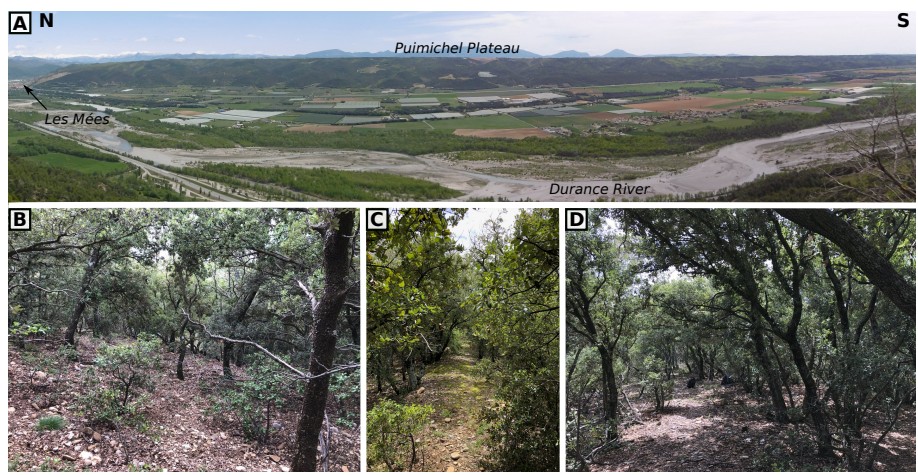

**Figure 3.** (A) Panorama of the Western edge of Puimichel Plateau and the studied basins, viewed from Ganagobie Abbey (see figure 1B for location). (B) Hillslope flank covered with regolith clasts. (C) Hilltop sampling site K (see figure 2A for location). (D) Hilltop sampling site M.

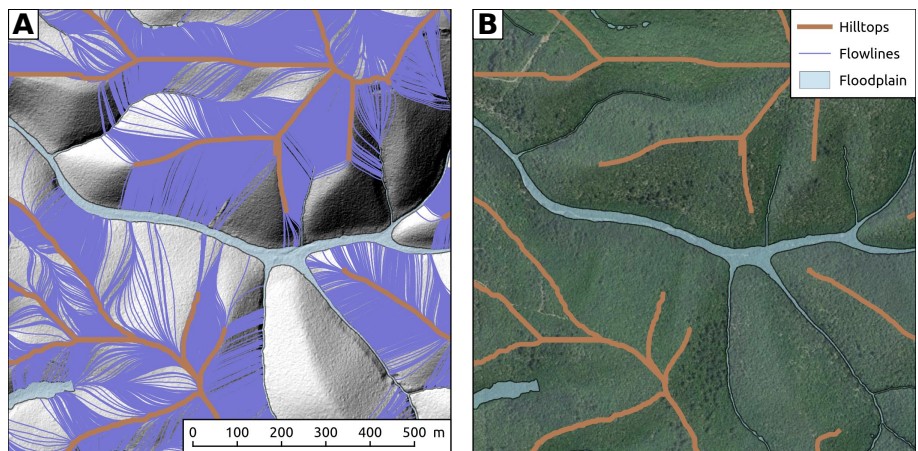

**Figure 4.** (A) Hillshade image from a 1 m IGN RGE ALTI® LiDAR Digital Terrain Model (DTM). Brown thick lines indicate the hilltops extracted from the DTM. Light blue polygon are the floodplains extracted using the approach of Clubb et al. (2017). Purple thin lines are flowlines routed from the hilltop toward the floodplain. (B) Corresponding orthophotography (IGN BD ORTHO®).

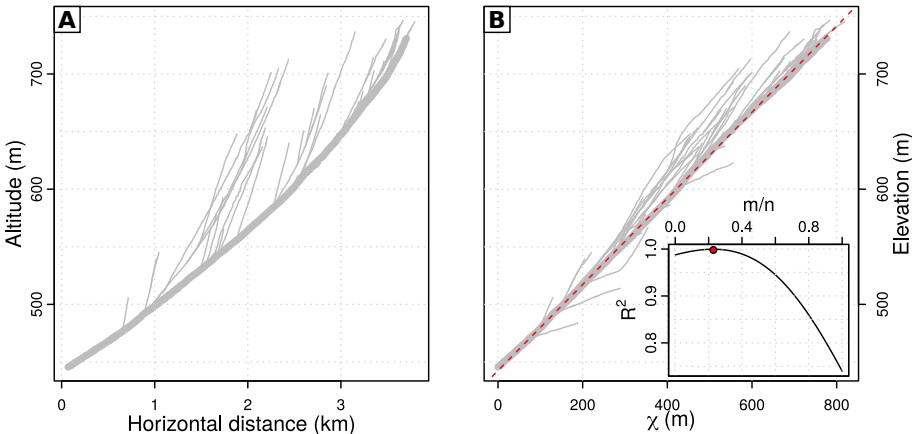

**Figure 5.** (A) River profile for the main trunk and tributaries of Moureisse catchment (see figure 2 for location). (B) $\chi$-transform of the longitudinal profile (Perron and Royden, 2012), using $m/n = 0.23$. Inset shows the evolution of the $R^2$ of the linear regression of elevation against $\chi$ for a range of $m/n$ values.

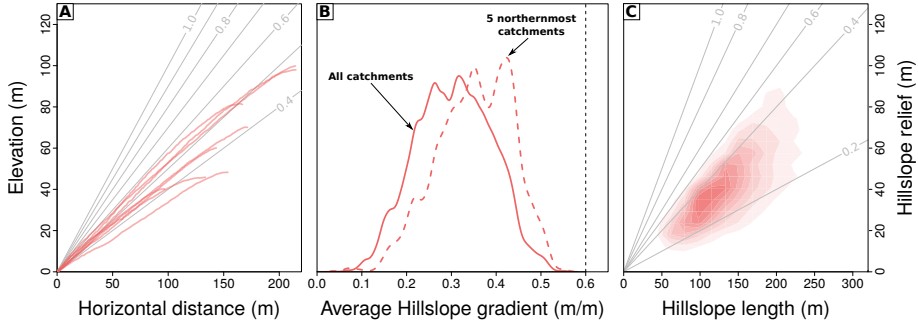

**Figure 6.** (A) Selected hillslope profiles from the studied basins (Figure 2). Grey lines indicate topographic gradient values. (B) Distributions of hillslope average topographic gradient for all the studied basins (solid line) and 5 northernmost basins (dashed line) where relief $R$, hilltop curvature $C_{HT}$ and non-dimensional erosion rate $E^*$ are the highest (Figure 7). (C) Joint distribution of hillslope relief and length for the flowlines extracted from the studied basins (Figure 4). Grey lines indicate topographic gradient values.

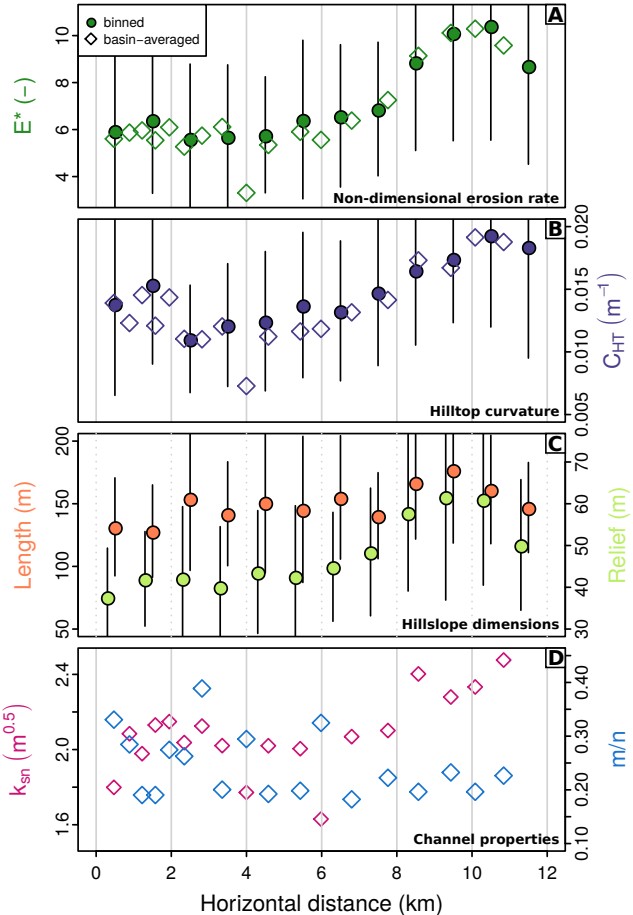

**Figure 7.** Projection of hillslope and fluvial parameters along the profile of figure 2. The horizontal distance values used here match the distance measured along the profile on figure 2. For all panels open symbols refer to basin averaged values (location of basins on figure 2) and closed symbols to 1-km length bin averages along the profile (error bars are $\pm 1\sigma$). A - Evolution of non-dimensional erosion rate, calculated as $E^* = 2L_H C_{HT}/S_c$ (Roering et al., 2007). B - Hilltop curvature computed over 15 m radius window. C - Hillslope length and relief from the flowlines patches (Figure 4). D - Normalized steepness index and $m/n$ ratio (concavity, reference value of 0.25) extracted from channel profiles of the studied basins.

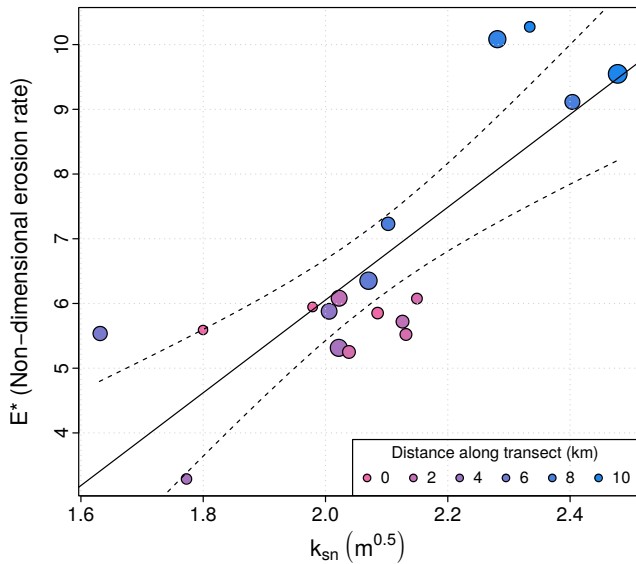

**Figure 8.** Evolution of basin-averaged $E^*$ as a function of normalized steepness index ($k_{sn}$). Circles are colored according to the position of the corresponding basins along the transect (Figures 2 and 7). Circle radius is a function of catchment size (ranging from 0.5 to 4 km$^2$). Solid and dashed lines correspond to a linear fit an its 95% confidence interval ($R^2 = 0.62$ and $p < 10^{-3}$).

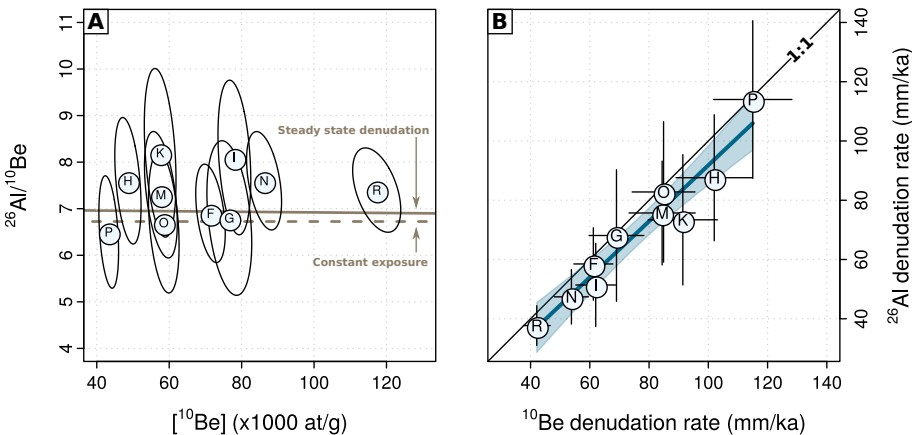

**Figure 9.** (A) Two nuclides plot for the sampled sites, with $1\sigma$ confidence ellipses (see figure 2 for location and tables 1 and 2 for data). Solid and dashed brown lines indicate the predicted $^{26}$Al/$^{10}$Be ratio for steady-state denudation and constant exposure histories, respectively. (B) Comparison of denudation rates calculated from measured $^{10}$Be and $^{26}$Al concentrations using the LSD scaling scheme (Lifton et al., 2014) and calculation procedure from Balco et al. (2008). Blue line and envelope are linear fit and its 95% confidence interval respectively.

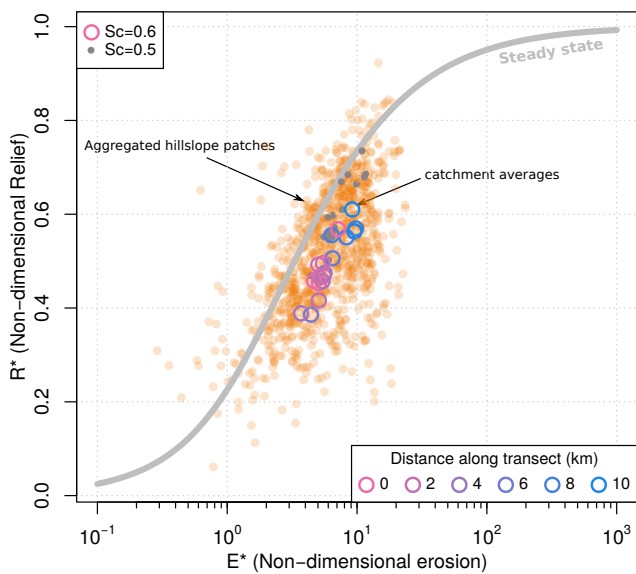

**Figure 10.** Basin-averaged $R^*$ vs $E^*$ plot (Roering et al., 2007; Grieve et al., 2016b). Two values for the critical hillslope gradient $S_c$ are tested. Open circles correspond to $S_c = 0.6$ m/m, colored according to the position of the basins along the transect (Figures 2 and 7). Small dark filled circles correspond to $S_c = 0.5$ m/m. Pale yellow symbols are averages over hilltop patches, for $S_c$=0.6 m/m (see text for details). Thick grey line corresponds to the steady state relationship between $R^*$ and $E^*$ predicted by equation 8.

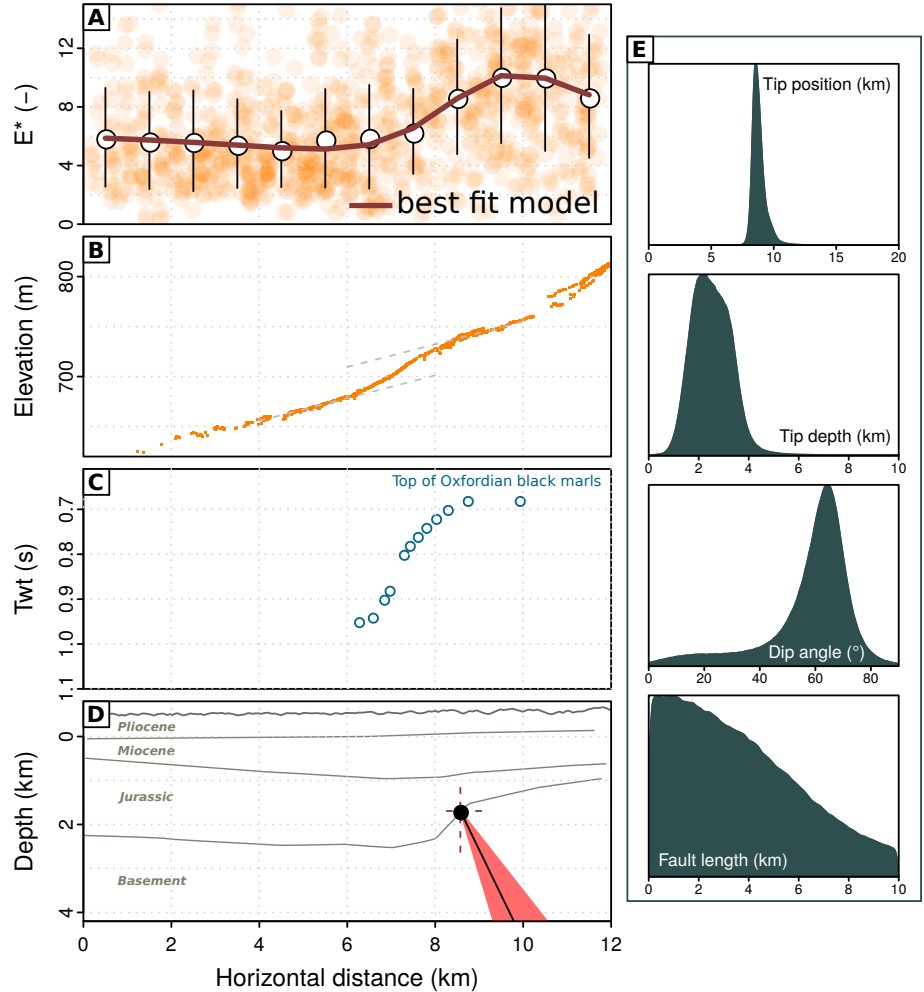

**Figure 11.** (A) Non-dimensional erosion rate evolution from hilltops patches (mean and standard deviation binned every 1 km). Red curve is the result of the optimization of a simple dislocation model (Okada, 1985), with amplitude adjusted to the range of $E^*$ values. See text for details. (B) Projection of the deformed summit surface of the Puimichel plateau (Figure 2B). (C) Two way travel times to the top of Oxfordian black marls, interpolated from seismic surveys across the studied area, as indicated in the report of drilling BSS002DWDJ available in the subsurface BSS database of BRGM. Data are projected on the section indicated as a dashed white line on figure 2C. (D) Geometry of the dislocation (dark red line) used to compute the surface deflection on panel A (red curve). Dashed lines (position and depth of the fault upper end) and light red surface (fault dip angle) correspond to 68% density intervals from the marginal distributions of panel E. Thin dark lines indicates the limits of the geological units from the cross section presented on figure 2D (Dubois and Curnelle, 1978). (E) Marginal distributions from a MCMC exploration for the parameters of the elastic dislocation model.

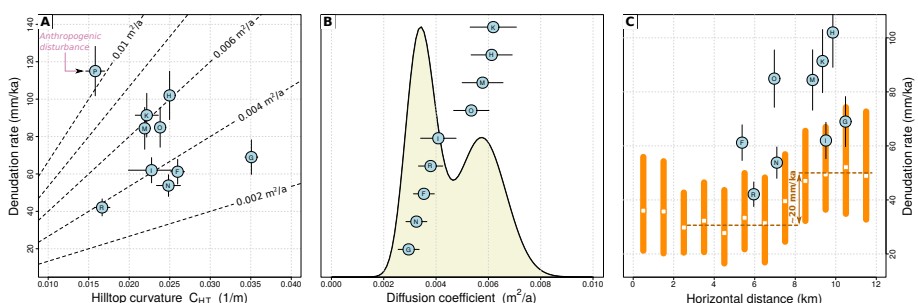

**Figure 12.** (A) [10]Be denudation rate against hilltop curvature. Dashed lines correspond to equation 3 for different values of the diffusion coefficient $D$. (B) Computed diffusion coefficient according to equation 3 at the various sites and corresponding probability density function. (C) Orange bars indicate the evolution of denudation rates along the profile from figure 7, calculated by sampling the distribution from the previous inset and applying equation 3 to individual hillslope patches. White squares are median values and orange bars indicate the interquartile range. Dashed horizontal lines delineate the $\sim$20 mm/ka differential denudation rate across the transect. For comparison, light blue circles indicate the denudation rates actually measured at hilltop sites (excluding site P). These rates are systematically higher than those computed from equation 3 and the evolution of $C_{HT}$ on figure 7B because sampling sites where selected on relatively high curvature ridges, higher than 0.02 m$^{-1}$ in most cases, whereas spatially averaged values along the transect are in average lower than 0.02 m$^{-1}$ (Figure 7B).

**Table 1.** Cosmogenic nuclides data : $^{10}$Be results

| Sample | Latitude ($^o$) | Longitude ($^o$) | Altitude (m) | Mass[a] (g) | Be carrier[b] (g) | $^{10}$Be/$^9$Be[c,d,e] ($\times 10^{-14}$) | [$^{10}$Be][d,f] (at/g) | $^{10}$Be denudation rate[d,g] (mm/ka) |
|--------|----------|-----------|----------|------|-----------|------------------|-----------|-------------------|
| TV-F | 43.9675 | 5.9476 | 482 | 18.88 | 0.1556 | $4.43 \pm 0.16$ | 71.36±2.67 | 61.2±6.6 |
| TV-G | 44.006 | 5.991 | 728 | 14.74 | 0.1518 | $3.83 \pm 0.2$ | 76.58±4.1 | 69±9.25 |
| TV-H | 44.0032 | 5.977 | 644 | 19.87 | 0.1523 | $3.29 \pm 0.15$ | 48.65±2.39 | 102±12.91 |
| TV-I | 44.0014 | 5.9707 | 614 | 18.24 | 0.1555 | $4.67 \pm 0.17$ | 77.99±2.95 | 62±6.73 |
| TV-K | 43.9964 | 5.9859 | 720 | 19.2 | 0.1522 | $3.74 \pm 0.18$ | 57.52±2.88 | 91.4±11.68 |
| TV-M | 43.9953 | 5.9705 | 616 | 16.38 | 0.1529 | $3.15 \pm 0.16$ | 57.68±3.05 | 84.4±11.18 |
| TV-N | 43.9805 | 5.9626 | 559 | 19.49 | 0.1455 | $5.81 \pm 0.21$ | 86.23±3.22 | 53.8±5.8 |
| TV-O | 43.9779 | 5.9684 | 644 | 18.58 | 0.1547 | $3.57 \pm 0.16$ | 58.48±2.82 | 84.9±10.61 |
| TV-P | 43.9748 | 5.971 | 644 | 19.65 | 0.1539 | $2.83 \pm 0.11$ | 43.28±1.82 | 115±13.21 |
| TV-R | 43.9687 | 5.9664 | 647 | 19.34 | 0.1408 | $8.07 \pm 0.29$ | 117.3±4.36 | 42.1±4.54 |

[a] Dissolved pure quartz mass.

[b] In-house carrier mass, ∼150 $\mu$l at 3.025 x $10^{-3}$ g/g (Merchel et al., 2008).

[c] $^{10}$Be/$^9$Be ratios were calibrated against the National Institute of Standards and Technology standard reference material 4325 by using an assigned value of 2.79±0.03×$10^{-11}$ (Nishiizumi et al., 2007).

[d] Uncertainties are reported at the $1\sigma$ level.

[e] Uncertainties on isotopic ratios are calculated according to the standard error propagation method using the quadratic sum of the relative errors and include a conservative 0.5% external machine uncertainty (Arnold et al., 2010), the uncertainty on the certified standard ratio, a $1\sigma$ uncertainty associated to the mean of the standard ratio measurements during the measurement cycles, a $1\sigma$ statistical error on counted events and the uncertainty associated with the chemical and analytical blank corrections.

[f] Two process blanks were treated and measured with our samples, yielding $^{10}$Be/$^9$Be ratios of 1.47±0.27 and 0.90±0.29 ×$10^{-15}$. It corresponds to an upper $1\sigma$ bound of 55 and 38 ×$10^3$ $^{10}$Be atoms for the background level in these two blanks, which is at least 20 times lower than the number of $^{10}$Be atoms in the dissolved sample masses (30 times lower in average over our dataset).

[g] Denudation rates were then calculated with the online calculator described in Balco et al. (2008) and the nuclide specific LSD scaling scheme (Lifton et al., 2014), using the CRONUS-Earth calibration dataset (Borchers et al., 2016) for the calculation of spallation production rates and muon production rates according to Balco (2017). We use 160 g/cm$^2$ for the effective attenuation length for spallation in rock, and a density of 2.5 g/cm$^3$. No shielding correction was considered for the hilltop sites we sampled, which were selected on nearly horizontal ridgelines.

**Table 2.** Cosmogenic nuclides data :[26]Al results and comparison with [10]Be results from table 1

| Sample | [Al][a] (ppm) | [26]Al/[27]Al[b,c] ($\times 10^{-13}$) | [[26]Al][c,d] (at/g) | [26]Al denudation rate[c,e] (mm/ka) | $\tau^{f}$ (ka) | [10]Be/[26]Al[c] | $\varepsilon_{10}/\varepsilon_{26}$[c] |
|---|---|---|---|---|---|---|---|
| TV-F | 173.6 | 1.33±0.1 | 489.13±41.9 | 58.5±12.23 | 10.5-10.9 | 6.9±0.7 | 1±0.2 |
| TV-G | 118.8 | 2.02±0.29 | 517.65±77.39 | 68.1±22.2 | 9.3-9.4 | 6.8±1.1 | 1±0.4 |
| TV-H | 136.8 | 1.29±0.12 | 371.72±39.07 | 87.6±21.28 | 6.3-7.3 | 7.6±0.9 | 1.2±0.3 |
| TV-I | 131.5 | 2.21±0.26 | 628.49±76.26 | 51.4±14.04 | 10.3-12.5 | 8.1±1.1 | 1.2±0.4 |
| TV-K | 151 | 1.47±0.19 | 471.82±64.22 | 73.4±22 | 7-8.7 | 8.2±1.2 | 1.2±0.4 |
| TV-M | 144.4 | 1.38±0.13 | 421.92±41.63 | 75.7±17.53 | 7.6-8.5 | 7.3±0.9 | 1.1±0.3 |
| TV-N | 139.6 | 2.16±0.15 | 651.3±49.63 | 47.4±9.18 | 11.9-13.5 | 7.6±0.7 | 1.1±0.3 |
| TV-O | 154.1 | 1.21±0.14 | 391.25±50.46 | 82.8±23.7 | 7.5-7.7 | 6.7±1 | 1±0.3 |
| TV-P | 149.8 | 0.91±0.08 | 280.9±28.33 | 114±26.6 | 5.6-5.6 | 6.5±0.8 | 1±0.3 |
| TV-R | 186.5 | 2.16±0.13 | 866.88±57.51 | 37.7±6.7 | 15.2-17 | 7.4±0.6 | 1.1±0.2 |

[a] Naturally occuring Al measured by ICP-OES. No Al carrier solution was added to the samples.

[b] The measured [26]Al/[27]Al ratios were normalized to the in-house standard SM-Al-11 whose ratio of $7.401\pm0.064\times10^{-12}$ (Arnold et al., 2010) has been cross-calibrated against primary standards from a round-robin exercise (Merchel and Bremser, 2004).

[c] Uncertainties are reported at the $1\sigma$ level.

[d] An analytical blank yielded a ratio of $7.27\pm2.17\times10^{-15}$.

[e] Procedures for that calculation of denudation rates are identical to the ones described in table 1.

[f] Integration time scales for [10]Be and [26]Al denudation rates von Blanckenburg (2005).