# Peer review of "Hillslope denudation and morphologic response to a rock uplift gradient"

_Earth Surface Dynamics, 2019_

## Referee Comment (RC1) · Martin D. Hurst (Referee) · 25 Nov 2019

The manuscript published for discussion by Godard et al. (2019) focuses on investigating the topographic signature of hillslopes and their variation with denudation rates in order to identify the spatial distribution of uplift and infer the structural control on this distribution via inverse modelling. The results tally well with a range of independent Geological observations. This is an ambitious and exciting piece of research that with some further clarification should be an excellent contribution to ESURF. I recommend publication subject to addressing the following comments and concerns:

Main comments

Apart from hillslopes that terminate at fault scarps or at the coast, it is channels that

set the base level conditions for hillslopes, and therefore hillslopes provide information about the history of channel erosion, not directly about uplift (without assuming or demonstrating a relationship between channels and uplift). The physical link between uplift, channels and hillslopes needs to be more clearly identified and discussed. I think expressing this more clearly throughout the manuscript will be beneficial as it has important implications for your results and interpretations. It seems paradoxical at first that hillslopes can record a denudation signal (and by inference, uplift) that is not also clearly identifiable in channel profiles. However, this serves to highlight that hillslopes may have a longer memory of past landscape development. So, while the channels may harbour little evidence of spatially varying uplift, the hillslope signal could provide evidence that some adjustment to uplift has previously taken place, and the hillslopes are still relaxing (as suggested by the E*R* relationship identified in your paper). Roering et al. (2001) developed an expression for the response timescale of nonlinear diffusion-like hillslopes that suggests response time is much longer for low erosion rates (and by inference, during relaxation), and this tallies with the relatively long decay signal we (Hurst et al., 2013) found at Dragon's Back Pressure Ridge. So I think you can say that hillslopes have the potential to record information about transient landscape dynamics either when channels have had time to adjust but hillslope morphology may still be responding.

The channel profile analysis is not presented with the same degree of rigour as was the case for the hillslope morphology in the methodology. I realise there has been a number of recent studies that have addressed the inference of tectonics from channel profile morphology but I think more explanation is needed relating to the approach to constraining theta and ksn. This is particularly the case as the value of theta chosen may affect whether there is a systematic variation in channel steepness. Could you directly compare channel and hillslope metrics on a catchment by catchment basis to show the presence or absence of a relationship?

The modelling approach used to identify structural control on rock uplift is poorly presented (despite a nice figure). Seems like an after-thought buried in the discussion but to my mind is one of the key novelties of the paper. I recommend that the modelling should be described in much more detail in the methods section and results presented and then discussed appropriately. This will require some restructuring of the paper but I think will lead to a more robust presentation.

Line-by-line comments and editorial corrections/suggestions

Title: suggest "response to a rock uplift gradient"

L2: replace "as" with "since"

L3: channels set baselevel for hillslopes, uplift may set baselevel for channels.

L6: cosmogenic nuclide-derived

L7-8: folds and thrusts plural

L9-10: CHT and non-dimensional

L10: delete "a" systematic. . .

L12: allows "us" to propose

L16: allows "us" to resolve

L22: singular "external forcing and tectonic. . ."

L24: highlight that channels set baselevel conditions for hillslopes

L26: unclear what "these two types if forcings" refers to. Restate tectonic and climatic here.

L32: tectonically spelling

L39: Duvall had some co-authors (et al.).

L44: Unclear what you mean by the "planar structure of river network", do you mean

planform?

L49-50: delete "rock uplift rates", hillslopes respond to channels

L53: typical lengths of hillslopes, could cite Grieve et al. (Grieve et al., 2016c) here since they investigated the variability of hillslope lengths in a variety of landscapes (as cited later on).

L58-60: Phrasing is a little unclear, could develop these ideas more logically/progressively.

L62: which can be used to infer patterns of uplift

L90: Plateau should be proper noun?

L93: Replace "At last", with "lastly"

L94: "Constrain" should be "constraints", plus comma needed at the end of this line.

L98: Figure should be capitalised? Or at least should be consistent throughout the paper, please check.

L110: Velensole Conglomeratic Formation all beginning with capitals

L159: Refer to Fig 2 here.

L162: qs has no units (units later provided on diffusion coefficient), need to be consistent.

L167: Derivation and presentation of Eq2 would make more sense if the Exner equation had been presented up front, rather than just referred to in words straight after.

L171: suggest you only need the Roering reference here.

L184-185: Cool! Would be great to make these tools available to the community.

L188: by "planar" I think you mean "planform"

L191: why this size of window? Might be worth looking at the (Grieve et al., 2016b) paper on grid resolution here. This paper is in your ref list but can't find cited in text. (Lashermes et al., 2007) were the first to consider this I think.

L193: Grieve paper year is 2016 not 2015.

L195-197: Less than three lines to describe channel profile analysis does not do it justice. Need to explain in more detail, for example what is Chi and why useful. How was concavity determined? Not as much detail as hillslope metrics perhaps needed but some more info required here.

L199: "allow US to identify..."

L200: suggest deleting "and under a steady-state assumption" since transient landscapes can also be used to interpret rock uplift distribution (though admittedly it's trickier).

L202: refer to Fig 2A here

L204: delete "a". Replace "drain" with "channel"

L235: density of soil is 2.5 (g/cm3 presumably?)? Seems high. Did you measure it?

L235: were hilltops plunging at all? Might require some adjustment if so.

L241: Present the panels in Figure 6 in order A-C or rearrange figure 6 to match the order you want to report in the text.

L243-244: Need reference to Figure 6A here. SC 0.6 needs justification. I would suggest 0.52 is quite a bit lower than values reported elsewhere. Can you look to previous studies for support. If your landscape doesn't contain hillslopes with mean slope approaching SC your estimates will be biased low.

L246: Fig 7A

L247: Add Fig 7B at end of line

L247-251: refer to figure and panels here

L255: Why was this reference value chosen? What are the mean/median values?

L258-259: Could this (null) result be sensitive to the choice of reference concavity? Would be worth checking.

L265: Fig 9 appears before Fig 8

L279-280: Can you present the data to demonstrate this lack of elevation change?

L283: No consideration of any potential variation in vegetation or landuse here (and not mentioned in the earlier section introducing the study site).

L313: replace "eventually" with "likely"

L314: could you colour code these points by distance along the transect?

L317-318: even dropping Sc this low all the average points sit below the steady state curve which only strengthens your claim.

L321-322: Again, concern about sensitivity relative to choice of m/n

L331-336: I would suggest that the key difference between the two is that along the Bolinas Ridge we see evidence for active channel transient response suggesting that channel adjustment is ongoing, unlike at your site.

L380-384: This paragraph is like a flash in the pan but there is a lot of important meat here. I would suggest that the modelling approach needs to be explained in full in the methods section and results presented in the results section, prior to discussion. What is MCMC acronym not defined? Which four parameters are free parameters? How was the ensemble set up to ensure full exploration of the parameter space? (flexibility of Markov Chain and acceptance rates). How quickly did the model converge on a solution? The results of this work are interesting and novel and deserve a more thorough treatment, which will require some restructuring.

L404-410: Repetition from intro.

L410: Acronym DTM has already been defined.

L424-425: (Grieve et al., 2016a) is perhaps an important reference here

L431: What evidence is there for human disturbance on the hilltops? Might this also have implications for hilltop curvature?

L435: What about vegetation as a control on D? Any spatial variation in vegetation?

L450: It seems to me that the absolute throw rate on a fault dipping at ∼60o should be larger than the vertical uplift (inferred from denudation) component, not the other way around. An uplift rate of 50mm/ka would imply a throw rate of ∼57mm/ka

L464: No discussion of vegetation gradient. Can you get above ground biomass from the LiDAR?

Figure 2: Panel A has no N arrow, and north arrow on panel B is hard to see.

Figure 3: Panel C might suggest some human influence on hilltops (humans have a habit of putting paths along hilltops).

Figure 4B: Cannot see flow lines above flood plain

Figure 6: reorder panels as per comments above

Figure 7: subscript on Ksn axis label. I would suggest sticking to either theta OR m/n, not both. Discussion of n values later would suggest the need for the latter. Stream power incision model has not been presented clearly.

Table 1 and 2: suggest following reporting recommendations of (Frankel, 2010).

References referred to in review:

Frankel, K. L.: Terrestrial Cosmogenic Nuclide Geochronology Data Reporting Standards Needed, Eos, Trans. Am. Geophys. Union, 91(4), 31–32,

doi:10.1029/2010EO040003, 2010.

Grieve, S. W. D., Mudd, S. M., Hurst, M. D. and Milodowski, D. T.: A nondimensional framework for exploring the relief structure of landscapes, Earth Surf. Dyn., 4(2), doi:10.5194/esurf-4-309-2016, 2016a.

Grieve, S. W. D., Mudd, S. M., Milodowski, D. T., Clubb, F. J. and Furbish, D. J.: How does grid-resolution modulate the topographic expression of geomorphic processes?, Earth Surf. Dyn., 4(3), 627–653, doi:10.5194/esurf-4-627-2016, 2016b.

Grieve, S. W. D., Mudd, S. M. and Hurst, M. D.: How long is a hillslope?, Earth Surf. Process. Landforms, doi:10.1002/esp.3884, 2016c.

Hurst, M. D., Mudd, S. M., Attal, M. and Hilley, G.: Hillslopes Record the Growth and Decay of Landscapes, Science (80-.  )., 341(6148), 868–871, doi:10.1126/science.1241791, 2013.

Lashermes, B., Foufoula-Georgiou, E. and Dietrich, W. E.: Channel network extraction from high resolution topography using wavelets, Geophys. Res. Lett., 34(23), doi:10.1029/2007GL031140, 2007.

Roering, J. J., Kirchner, J. W. and Dietrich, W. E.: Hillslope evolution by nonlinear, slope-dependent transport: Steady state morphology and equilibrium adjustment timescales, J. Geophys. Res., 106(B11), 26787, doi:10.1029/2001JB900018, 2001.

---

## Referee Comment (RC2) · Marta Della Seta (Referee) · 9 Dec 2019

In the manuscript entitled " Hillslope denudation and morphologic response across a rock uplift gradient" Godard et al. present the results of a very interesting methodological test focused on high resolution analysis of hillslope morphology as proxy for uplift gradients. The analyses are supported by independent geological and geochronological data and the results of this research undoubtedly contribute to outline the potential of the very dense information associated to hillslope morphometry from high resolution DTMs to record tectonic rates. The manuscript is well-written, overall well-structured and I recommend publishing it in Earth Surface Dynamics, after minor revisions, according to the following general and line-by-line comments:

GENERAL COMMENTS:

1. The interpretation of the factors controlling hillslope morphology (I suggest using "morphometry") is certainly supported by strong independent geological constraints on deep structures responsible to variable surface uplift along the transect used for the analyses. Nonetheless, this part of the discussions should clarify better the complete set of factors controlling the hillslope morphometry. For example, despite the climatic and lithological homogeneity, could the catchment size/shape play a role in the results obtained? Given that I'm convinced that high resolution DTMs are incredible sources of morphometric data to be interpreted in the light of the theoretical landscape evolution (Vergari et al., 2019, DOI: 10.1002/esp.4496), the high resolution itself imply that your DTM records very local surface features: despite the binning, how can you exclude that the results of your morphometric analyses along the transect are affected by the occurrence of local processes related to the hillslope-channel dynamics?

2. Section 5.3 shows new data coming from the application of a dislocation model to predict surface deformations associated to tectonic structures. Therefore, I suggest moving it to the results, after having better explained the method you used.

3. If I did not misunderstand, your inferences about the lacking record of the uplift gradient in the river long profiles is based on the analysis of all the catchments, so why do you show the profiles and CHI-plot just for a single catchment (fig. 5)? In my opinion this part should be better presented in order to strengthen the implications of your results, which concern the transient conditions recorded in the hillslope morphometry when the main trunks and tributary channels already adjusted to tectonic perturbations (in case of low tectonic rates/relatively old tectonic input). I suggest you refer also to the papers by Demoulin (2011 doi:10.1016/j.geomorph.2010.10.033, 2012 doi:10.1029/2012GL052201), who based his landscape metrics on the diachronic response to tectonic perturbations by main trunks, tributaries and hillslopes within catchments.

4. In the discussions you state that "In the absence of any climatic, litho- logic or vegetation gradient, the observed increase in hilltop curvature, hillslope relief and nor- malized erosion rate points to a coincident increase in rock uplift". This seems to be a weakness of the method, since often such a homogeneity of climatic, lithologic and vegetation cover factors is lacking, especially when dealing with areas with regional extent. Maybe you'd better discuss this point before the conclusions.

5. Some figure are too small to be readable (Figs 1, 6, 11).

LINE-BY-LINE COMMENTS:

Title: I suggest changing "morphologic" with "morphometric"

L13: maybe you mean "eroded" conglomerate

L26: types of forcing

L42-45: references for CHI maybe deserve to be cited

L70: correct "hilltope" in "hilltop"

L129: Myr, not M.yr. Moreover, check the uppercase for East and West (uppercase not used in other sentences, please homogeneize)

L131: Lambruissier not readable in Fig. 1

L151-153: The reason why the hillslopes can be considered as regolith-mantled and transport-limited systems is not so clear to me.

L165: I'm not sure that "shallow" is the right adjective for slope.

L246: South to North, maybe?

L252-254: What stated is not so clear in Fig. 5: first of all because here are reported data only from a single catchment; secondly, not all the profiles shown have a concave- up shape (linear CHI-transformed shape).

L256: South to North, maybe?

L265: Figure 9 is cited before Figure 8, maybe better inverting their numbering.

L265-266: The maximum denudation rate obtained from 26Al as declared in the text does not fit with the data plotted in Fig. 9B (here, sample P seems showing maximum denudation rate obtained from 26Al >88 mm/ka).

L270: You should motivate the choice of considering only 10Be.

L279: what do you mean with "important" when referring to range in elevation of the catchments from South to North?

L307: Maybe "The uplift itself is due to a long..."

L324: E* and CHT do not show a gradual increase as stated, rather an abrupt increase.

L341-341: maybe the lack of a correlation between LH and CHT differently from Hurst et al. 2013 could depend on the fact that in the submitted manuscript the metrics are measures along a narrow profile transverse to divides.

L356: does not depend

L375: you'd better discuss the assumption of a single planar dislocation in an elastic medium used in tour modelling

L393: formations underwent long-wavelength

L413: ...providing the framework

L431: did you evaluate the possibility of other disturbances in your catchments before performing CRN analyses?

L478: maybe comparison instead of confrontation
* * *

---

## Referee Comment (RC3) · Peter van der Beek (Referee) · 14 Dec 2019

General comments:

Godard et al. exploit a high-resolution Lidar-derived digital terrain model to explore the hill-slope response to a rock-uplift gradient in a well constrained, climatically and lithologically homogeneous setting: the Valensole Plateau in SE France. In particular, they show that systematic variations in ridge curvature and a derived non-dimensional erosion rate can be linked to subtle tectonic uplift of the northern part of the plateau, which they proceed to model with an elastic-dislocation model as resulting from compressional reactivation of a pre-existing normal fault in the basement underlying the plateau. Finally, they use cosmogenic 10Be and 26Al measurements of ridge erosion

rates to dimensionalise the analysis and provide estimates of the rates of shortening, uplift and erosion of the study area.

This manuscript will make an excellent contribution to ESurf. It shows how high-resolution DTM data can be used in conjunction with cosmogenic nuclide data to provide detailed inferences about landscape response to variable uplift rates and shows a way forward in morphotectonic analysis. The analysis is very complete, as the authors assess both hill-slope and channel response, include cosmogenic nuclide data to obtain actual rates, and interpret the results in terms of a tectonic driver using a numerical model inversion. It is therefore timely and of broad interest to the ESurf readership. It is also (mostly) well written and illustrated. I therefore recommend this be accepted pending minor/moderate revision.

Specific comments:

There are a few issues the authors could address in more detail in a revision:

First, the uplift gradient discussed in the manuscript is superimposed on a longer-wavelength uplift gradient that is clearly recorded in both present-day vertical motions as recorded by permanent GPS stations (e.g. Noquet et al., 2016), and deformed geomorphic surfaces such as the Valensole plateau and the alluvial terraces of the Durance River (e.g. Champagnac et al., 2008). While this is noted in passing in the manuscript, the tilting of the Valensole plateau would lead to increasing uplift/erosion rates toward the northeast even in the absence of an active fault. It could be made clearer in the manuscript that the authors are investigating a shorter-wavelength uplift pattern superimposed on the regional pattern. In keeping with this, if the interpretation of the authors is correct, the Durance river terraces, acting as passive markers, should also record the fault offset. Whereas the long-wavelength tilting is recorded by these terraces (e.g. Champagnac et al., 2008 and references therein) it is not clear whether the faulting is. The terraces are reported to be up to 1 Ma old, i.e. up to about half the age of the surface of the Valensole plateau, and should therefore record up to half

the offset of the surface (estimated at ∼30 m from Figs. 2 and 10), largely sufficient to be recorded by the high-resolution DTM. This aspect could be looked at, and the implications of the surface geometry of the terraces discussed, in some more detail.

The analysis of the river profiles implicitly assumes they behave as detachment-limited bedrock streams and that they have reached steady state with respect to uplift. Neither assumption is immediately obvious: given that these streams incise relatively non-consolidated coarse conglomerates, they could very well behave as transport-limited streams; and given the probably recent onset of faulting below the Valensole plateau, they may not have equilibrated to the uplift regime yet. Therefore, the fact that the stream-profile analysis led to less clear constraints than the hill-slope analysis could be due to inadequate theory as much as insufficient resolution. Again, this aspect could be discussed in some more detail.

The discussion regarding whether the hill slopes have reached steady state (Fig 8) is not very clear. Some explanation of why R* and E* should vary as indicated in Fig. 8 in case of steady state appears to be lacking.

Finally, the procedure to extrapolate locally measured erosion rates (Fig. 11) appears a little convoluted. I understand the authors' rationale for doing this, as it is not obvious in how much the measured erosion rates are representative, but the resulting pattern could be perceived as being somewhat removed from the actual data. A potential solution to this could be to simply plot the measured erosion rates at the corresponding locations in Fig. 11, so that it is easy to assess in how much the followed procedure modifies these measured rates.

Technical corrections:

Although the writing is overall of good quality, a number of minor English-language issues distract somewhat from the science. These include frequent singular/plural confusions, illogical comma use and awkward sentence structures (placement of adjectives, etc.). I am returning an annotated manuscript directly to the authors so they can correct

these minor issues.

Figures are mostly clear and well drafted. I would suggest use of another colour than yellow for symbols or profile lines, as I found these very hard to read (e.g. in Fig. 2B, 8, 10A and B). Figure 2 A and C need a (larger) N-arrow to indicate they are rotated nearly 90°; labelling the Durance River would also help.

---

## Editor Comment (EC1) · Veerle Vanacker (Editor) · 14 Jan 2020

Dear authors,

By now, we have received three reviews of your paper. The three reviewers agree on the high quality of the presented work, and they particularly appreciated your integrated work combining high-resolution topographic analyses, CRN data and numerical modelling.

The reviewers recommend moderate revisions, and I consider the following issues to be the most critical ones:

(1) While the hillslope analyses are clearly described in the methods' section of the manuscript, this is less so for the stream channel analysis and modelling approach. I

refer here to the comments of Reviewer#1, and comments 2 and 3 of Reviewer#2 that will require further elaboration of methodological aspects.

(2) In the paper, you mention that the "hillslope analysis allows to resolve variations in rock uplift, that would not be possible to resolve using stream profile analysis". The three reviewers posted some critical notes here. Reviewer#1 makes the remark that hillslopes might have a longer memory than stream channels because of the differences in response timescale. A related comment is made by Reviewer#2 who points to the possibility of having diachronic response to tectonic perturbations within larger watersheds. Reviewer#3 asks for a discussion of the assumptions of the stream profile analyses that are designed for detachment-limited bedrock streams that are in topographic steady state with respect to rock uplift.

(3) Reviewer#3 suggests to clarify that this study mainly addresses "the short-wavelength uplift patterns that are superimposed on the regional pattern". This element can be further elaborated in the discussion.

More details are given in the reviewers' comments. Please document the modifications in term of content and line numbers.

Looking forward to receive your revised manuscript.

---

## Author Comment (AC1) · 18 Jan 2020

Dear Pr. Vanacker,

Please find our response (as an attached supplement) to reviewers comments on the manuscript entitled Hillslope denudation and morphologic response to a rock uplift gradient (esurf-2019-50), on behalf of myself and co-authors.

We thank all 3 reviewers for their highly valuable and constructive comments on our manuscript. Based on these reviews the main changes are the following.

- We have clarified an important point concerning the response of the fluvial network to the rock uplift gradient : we do observe a response, but the pattern is less resolved than what we obtain with the hillslope-based metrics. A new figure (8) has been added

to better highlight this point (correlation between ksn and E*).

- We have reorganized and expanded the Methods section (presentation of the river profile analysis and the modelling approach).

- We have reorganized the structure of the discussion (merging of the previous 5.1 and 5.3 subsections, displacement of some of the 5.3 subsection into the Methods).

- The data table have been expanded and reorganized into two different tables.

We will provide an edited version of the article where important additions and modifications are highlighted in red. We have complied with all the minor comments and suggestions relative to improving the text (spelling, formulations, typos) that were kindly provided by the reviewers, so we do not elaborate on the resulting modifications hereinafter, and, for the sake of readability, these small edits are not highlighted in the revised version of the manuscript.

Best regards,

Vincent Godard

Please also note the supplement to this comment:
https://www.earth-surf-dynam-discuss.net/esurf-2019-50/esurf-2019-50-AC1-supplement.pdf

**Supplement:**

Vincent GODARD Associate Professor Aix-Marseille Université CEREGE godard@cerege.fr

Aix-en-Provence, January 14th 2020

Pr. V. Vanacker Editor Earth Surface Dynamics

Dear Pr. Vanacker,

Please find our response to reviewers comments on the manuscript entitled *Hillslope denudation and morphologic response to a rock uplift gradient* (esurf-2019-50), on behalf of myself and co-authors.

We thank all 3 reviewers for their highly valuable and constructive comments on our manuscript. Based on these reviews the main changes are the following.

- We have clarified an important point concerning the response of the fluvial network to the rock uplift gradient : we do observe a response, but the pattern is less resolved than what we obtain with the hillslope-based metrics. A new figure (8) has been added to better highlight this point (correlation between  $k_{sn}$  and  $E^*$ ).
- We have reorganized and expanded the *Methods* section (presentation of the river profile analysis and the modelling approach).
- We have reorganized the structure of the discussion (merging of the previous 5.1 and 5.3 subsections, displacement of some of the 5.3 subsection into the *Methods*).
- The data table have been expanded and reorganized into two different tables.

We will provide an edited version of the article where important additions and modifications are highlighted in red. We have complied with all the minor comments and suggestions relative to improving the text (spelling, formulations, typos) that were kindly provided by the reviewers, so we do not elaborate on the resulting modifications hereinafter, and, for the sake of readability, these small edits are not highlighted in the revised version of the manuscript.

Best regards,

Vincent Godard

V. GODAKE

**Editor (Veerle Vanacker)**

Dear authors,

By now, we have received three reviews of your paper. The three reviewers agree on the high quality of the presented work, and they particularly appreciated your integrated work combining high-resolution topographic analyses, CRN data and numerical modelling. The reviewers recommend moderate revisions, and I consider the following issues to be the most critical ones:

(1) While the hillslope analyses are clearly described in the methods' section of the manuscript, this is less so for the stream channel analysis and modelling approach. I refer here to the comments of Reviewer#1, and comments 2 and 3 of Reviewer#2 that will require further elaboration of methodological aspects.

As described in the response to reviewers 1 and 2 a detailed presentation of the stream profile analysis is now provided in the *Methods* section and the presentation of the modelling approach has been expanded and moved to the *Methods* section.

(2) In the paper, you mention that the "hillslope analysis allows to resolve variations in rock uplift, that would not be possible to resolve using stream profile analysis". The three reviewers posted some critical notes here. Reviewer#1 makes the remark that hillslopes might have a longer memory than stream channels because of the differences in response timescale. A related comment is made by Reviewer#2 who points to the possibility of having diachronic response to tectonic perturbations within larger watersheds. Reviewer#3 asks for a discussion of the assumptions of the stream profile analyses that are designed for detachment-limited bedrock streams that are in topographic steady state with respect to rock uplift.

This aspect required indeed some clarification, as our initial presentation was perhaps ambiguous. As explained in our response to the reviewer below our main point is that, for short-wavelength variations in tectonic forcing, the far higher density of information offered by hillslope analysis allows a better sampling of the underlying rock uplift pattern. The change in rock uplift along our transect is actually observable with our  $k_{sn}$  measurements on figure 7. We have added a new figure 8, which allows to observed the existence of a significant correlation between fluvial ( $k_{sn}$ ) and hillslope (E\*) -derived proxies for rock uplift. The text has been edited to remove this initial ambiguity.

(3) Reviewer#3 suggests to clarify that this study mainly addresses "the shortwavelength uplift patterns that are superimposed on the regional pattern". This element can be further elaborated in the discussion. More details are given in the reviewers' comments.

This difference between the short- and long-wavelength deformation patterns has been highlighted in more details at several places in the text.

Please document the modifications in term of content and line numbers. Looking forward to receive your revised manuscript.

**Reviewer #1 Martin D. Hurst**

The manuscript published for discussion by Godard et al. (2019) focuses on investigating the topographic signature of hillslopes and their variation with denudation rates in order to identify the spatial distribution of uplift and infer the structural control on this distribution via inverse modelling. The results tally well with a range of independent Geological observations. This is an ambitious and exciting piece of research that with some further clarification should be an excellent contribution to ESURF. I recommend publication subject to addressing the following comments and concerns:

**Main comments**

Apart from hillslopes that terminate at fault scarps or at the coast, it is channels that set the base level conditions for hillslopes, and therefore hillslopes provide information about the history of channel erosion, not directly about uplift (without assuming or demonstrating a relationship between channels and uplift). The physical link between uplift, channels and hillslopes needs to be more clearly identified and discussed. I think expressing this more clearly throughout the manuscript will be beneficial as it has important implications for your results and interpretations. It seems paradoxical at first that hillslopes can record a denudation signal (and by inference, uplift) that is not also clearly identifiable in channel profiles. However, this serves to highlight that hillslopes may have a longer memory of past landscape development. So, while the channels may harbour little evidence of spatially varying uplift, the hillslope signal could provide evidence that some adjustment to uplift has previously taken place, and the hillslopes are still relaxing (as suggested by the E\*R\* relationship identified in your paper). Roering et al. (2001) developed an expression for the response timescale of nonlinear diffusion-like hillslopes that suggests response time is much longer for low erosion rates (and by inference, during relaxation), and this tallies with the relatively long decay signal we (Hurst et al., 2013) found at Dragon's Back Pressure Ridge, So I think you can say that hillslopes have the potential to record information about transient landscape dynamics either when channels have had time to adjust but hillslope morphology may still be responding.

Right. We think that there was some ambiguity in the initial presentation of our results. As our focus throughout the manuscript is on hillslope morphology, we probably did not invest enough time in the presentation of the methods and description of the results derived from river profiles morphology (as rightly pointed out by the reviewer in the next comment), and this might yield the misleading impression of a lack of response of these river profiles. Actually, as was briefly mentioned in the initial ms, figure 7D shows that  $k_{sn}$  values are higher in the northern part of the transect, but do not show as clear a progressive evolution as the hillslope-derived metrics. The overall message we want to convey is that both systems are responding to spatial changes in rock uplift, but the short wavelength variation, associated with a single structure, which occurs along our transect are much better captured by hillslope metrics, due to the orders of magnitude difference in the density of information they provide. We have made several edits throughout the text to clarify this point (see new figure 8 for example).

The channel profile analysis is not presented with the same degree of rigour as was the case for the hillslope morphology in the methodology. I realise there has been a number of recent studies that have addressed the inference of tectonics from channel profile morphology but I think more explanation is needed relating to the approach to constraining theta and ksn. This is particularly the case as the value of theta chosen may affect whether there is a systematic variation in channel steepness. Could you directly compare channel and hillslope metrics on a catchment by catchment basis to show the presence or absence of a relationship?

We have expanded the *Methods* part to provide a better presentation of the fluvial metrics used here. We have also added a new figure with a scatter plot of  $E^*$  as a function of  $k_{sn}$ , showing that these two metrics are significantly correlated. In line with previous comment, we realize that our presentation of the  $k_{sn}$  data was initially ambiguous, we do not claim that  $k_{sn}$  does not display any variations along our transect (the northern basins display higher values as initially stated), but that there is an important scatter in the such data, making the analysis of the pattern difficult, whereas the much higher data density distribution of the hillslope metrics yield a much more resolved pattern. We have modified our presentation of the data to make this point more apparent. We have also slightly changed the reference value of theta. While it obviously changed the absolute  $k_{sn}$  values, it had no influence on their relative distribution.

The modelling approach used to identify structural control on rock uplift is poorly presented (despite a nice figure). Seems like an after-thought buried in the discussion but to my mind is one of the key novelties of the paper. I recommend that the modelling should be described in much more detail in the methods section and results presented and then discussed appropriately. This will require some restructuring of the paper but I think will lead to a more robust presentation

We have moved and expanded the description of the approach we use to interpret the E\* pattern into the *Methods* section (*Surface deformation modelling* subsubsection). However, we kept most of the corresponding results in the discussion as moving the outcome of the modeling into the *Result/Data* section does not seems appropriate for the following reasons:

- the first part of the discussion is devoted to argument in favor of a tectonic origin for the observed pattern and to weigh the merits of various alternative hypotheses. As the most important assumption behind the model is that the surface erosion patterns are driven by differences in tectonic rock-uplift, presenting the modeling results earlier would result in an awkward inversion in the progression of the ms.
- The *Results* section has been renamed *Data*, in order to better reflect that it is devoted to the objective presentation of the primary observations made during this study (topographic metrics and CRN-derived denudation rates), with no interpretation at this stage. The modelling results clearly do not belong here.

We have restructured the discussion, with a first subsection incorporating subsections 5.1 and 5.3 of the original ms. We also reran the whole MCMC analysis after a slight correction to the definition of our likelihood function, yielding narrower marginal pdfs.

Line-by-line comments and editorial corrections/suggestions

Title: suggest "response to a rock uplift gradient" Done

L2: replace "as" with "since" Done

L3: channels set baselevel for hillslopes, uplift may set baselevel for channels. Done

L6: cosmogenic nuclide-derived

**Done**

L7-8: folds and thrusts plural Done

L9-10: CHT and non-dimensional Done

L10: delete "a" systematic... Done

L12: allows "us" to propose Done

L16: allows "us" to resolve Done

L22: singular "external forcing and tectonic..." Done

L24: highlight that channels set baselevel conditions for hillslopes Done

L26: unclear what "these two types if forcings" refers to. Restate tectonic and climatic here.

**Done**

L32: tectonically spelling Done

L39: Duvall had some co-authors (et al.). Done

L44: Unclear what you mean by the "planar structure of river network", do you mean planform?

**Done**

L49-50: delete "rock uplift rates", hillslopes respond to channels Done

L53: typical lengths of hillslopes, could cite Grieve et al. (Grieve et al., 2016c) heresince they investigated the variability of hillslope lengths in a variety of landscapes (ascited later on).

**Done**

L58-60:Phrasing is a little unclear, could develop these ideas more logically/progressively.

**Done. Sentence reorganized and simplified.**

L62: which can be used to infer patterns of uplift Done

L90: Plateau should be proper noun? Done

L93: Replace "At last", with "lastly" Done

L94: "Constrain" should be "constraints", plus comma needed at the end of this line. Done

L98: Figure should be capitalised? Or at least should be consistent throughout thepaper, please check.

**Done**

L110: Velensole Conglomeratic Formation all beginning with capitals Done

L159: Refer to Fig 2 here. Done

L162: qs has no units (units later provided on diffusion coefficient), need to be consistent.

**Done**

L167: Derivation and presentation of Eq2 would make more sense if the Exner equation had been presented up front, rather than just referred to in words straight after. Done

L171: suggest you only need the Roering reference here. Done

L184-185: Cool! Would be great to make these tools available to the community. Some of these are almost packaged as Grass modules and we plan to put them on the Grass add-ons page, but we need to simplify the input/output and write proper documentation before.

**L188: by "planar" I think you mean "planform" Done. Changed to contour curvature**

L191: why this size of window? Might be worth looking at the (Grieve et al., 2016b)paper on grid resolution here. This paper is in your ref list but can't find cited in text.(Lashermes et al., 2007) were the first to consider this I think. Done. Citation added and text amended.

L193: Grieve paper year is 2016 not 2015. Done

L195-197: Less than three lines to describe channel profile analysis does not do it justice. Need to explain in more detail, for example what is Chi and why useful. Howwas concavity determined? Not as much detail as hillslope metrics perhaps neededbut some more info required here.

Done, a dedicated subsubsection in the Methods is now devoted to the presentation of the underlying theory (see reply to main comment above).

L199: "allow US to identify..." Done L200: suggest deleting "and under a steady-state assumption" since transient landscapes can also be used to interpret rock uplift distribution (though admittedly it's trick-ier).

**Done**

L202: refer to Fig 2A here Done

L204: delete "a". Replace "drain" with "channel" Done

L235: density of soil is 2.5 (g/cm3 presumably?)? Seems high. Did you measure it? It is indeed higher than typical soil. We did not measure the density, but use this high value, because the regolith is mostly constituted of sandstone and limestone clasts.

L235: were hilltops plunging at all? Might require some adjustment if so. No, one of the main criteria used when selecting the sampling site was for them to be located along near horizontal ridgelines (information added).

L241: Present the panels in Figure 6 in order A-C or rearrange figure 6 to match the order you want to report in the text.

Done. Insets of the figure reordered.

L243-244: Need reference to Figure 6A here. SC 0.6 needs justification. I would suggest 0.52 is quite a bit lower than values reported elsewhere. Can you look to previous studies for support. If your landscape doesn't contain hillslopes with mean slope approaching SC your estimates will be biased low.

Done. Further justification for the value of  $S_c$  added in 4.1 (specific nature of the regolith made of cobble sized clasts with little extra cohesion added by sparse vegetation). Northern most catchments display almost linear hillslopes over more than half of their length.

L246: Fig 7A Done

L247: Add Fig 7B at end of line Done

L247-251: refer to figure and panels here Done

L255: Why was this reference value chosen? What are the mean/median values? The reference value was changed to 0.25 (average value = 0.24), this is now indicated in the text.

L258-259: Could this (null) result be sensitive to the choice of reference concavity? Would be worth checking.

The initial presentation of the  $k_{sn}$  results was awkward, there is indeed a slight increase along the transect (northern values higher), which is blurred by the scatter in the data. We have modified this in the text and also introduced a new figure showing a positive correlation between E\* and ksn. The relative values of  $k_{sn}$  (higher at the northern end of the transect) were not affected by the change from 0.3 to 0.25.

L265: Fig 9 appears before Fig 8 Done

L279-280: Can you present the data to demonstrate this lack of elevation change? Done. Average elevation and relief of catchments indicated.

L283: No consideration of any potential variation in vegetation or land use here (and not mentioned in the earlier section introducing the study site). Done. Information about homogeneous vegetation provided.

L313: replace "eventually" with "likely" Done

L314: could you colour code these points by distance along the transect? Done

L317-318: even dropping Sc this low all the average points sit below the steady state curve which only strengthens your claim.

We are not sure the slight deviation from the steady state curve can be considered significant in this case.

L321-322: Again, concern about sensitivity relative to choice of m/n See response to main comment above. We tested another value of m/n, which did

modify the absolute  $k_{sn}$  values but not the relative variation.

L331-336: I would suggest that the key difference between the two is that along the Bolinas Ridge we see evidence for active channel transient response suggesting that channel adjustment is ongoing, unlike at your site. **Done**.

L380-384: This paragraph is like a flash in the pan but there is a lot of important meat here. I would suggest that the modelling approach needs to be explained in full in the methods section and results presented in the results section, prior to discussion. What is MCMC acronym not defined? Which four parameters are free parameters? How was the ensemble set up to ensure full exploration of the parameter space? (flexibility of Markov Chain and acceptance rates). How quickly did the model converge on a solution? The results of this work are interesting and novel and deserve a more thorough treatment, which will require some restructuring.

Done, see reply to main comment above. A specific subsection in the *Methods* is now dedicated to the presentation of the approach.

**L404-410: Repetition from intro.**

Right, but this is one of the key idea (combination of high-res morphological analysis with geochronology data) that we want to put forward in this ms, and we feel it is important to come back to it here.

L410: Acronym DTM has already been defined. DEM? Done

L424-425: (Grieve et al., 2016a) is perhaps an important reference here There is no explicit comparison of CRN data with topographic metrics in this paper?

L431: What evidence is there for human disturbance on the hilltops? Might this also have implications for hilltop curvature?

Done, information added to the main text. There were small cobble-made walls close to the sampling site, and in some part shallow deflection of the surface, which were possibly due to excavation (and thus exhumation of lower concentration material). the typical wavelength was <1m and they are unlikely to affect the measurements of curvature done here.

L435: What about vegetation as a control on D? Any spatial variation in vegetation? Done. Vegetation cover is homogeneous along the studied transect, this is now indicated in the first part of the discussion.

L450: It seems to me that the absolute throw rate on a fault dipping at~600 should be larger than the vertical uplift (inferred from denudation) component, not the other wayaround. An uplift rate of 50mm/ka would imply a throw rate of~57mm/ka

That's what we did, but we are using the differential uplift of 20 mm/ka between the S and N parts of the transect as an estimate of the vertical displacement across the fault (corresponding figure modified for clarity). Converting into slip rates give 23 mm/yr, given the large uncertainties at this stage of the calculation, we report that as ~20 mm/ka in the text. The text has been edited for clarification.

L464: No discussion of vegetation gradient. Can you get above ground biomass from the LiDAR?

Vegetation cover is now mentioned in the discussion. We only have access to the DSM built from last returns (IGN RGE national grid), and no way to assess vegetation height from these data.

Figure 2: Panel A has no N arrow, and north arrow on panel B is hard to see. Done

Figure 3: Panel C might suggest some human influence on hilltops (humans have a habit of putting paths along hilltops).

The axis of the ridge (where the sampling site is located) is actually slightly off to the left (and on the right on panel D). We tried as much as possible to select such configuration for our sampling sites. The small paths that sometimes follow the hilltops did not show any sign of significant excavation or mobilization of sediments.

Figure 4B: Cannot see flow lines above flood plain

We stopped the flowlines at the edge of the floodplain (information added to methods section)

Figure 6: reorder panels as per comments above Done

Figure 7: subscript on Ksn axis label. I would suggest sticking to either theta OR m/n,not both. Discussion of n values later would suggest the need for the latter. Stream power incision model has not been presented clearly.

Done. A new subsusbection in methods is dedicated to the presentation of the river profiles analysis.

Table 1 and 2: suggest following reporting recommendations of (Frankel, 2010) Done. The presentation of the results has been organized into 2 tables for the sake of readability. Most of the technical information, initially presented in the text, has been moved to the tables captions for easier reference.

**Reviewer #2 Marta Della Seta**

In the manuscript entitled "Hillslope denudation and morphologic response across a rock uplift gradient" Godard et al. present the results of a very interesting methodological test focused on high resolution analysis of hillslope morphology as proxy for uplift gradients. The analyses are supported by independent geological and geochronological data and the results of this research undoubtedly contribute to outline the potential of the very dense information associated to hillslope morphometry from high resolution DTMs to record tectonic rates. The manuscript is well-written, overall well-structured and I recommend publishing it in Earth Surface Dynamics, after minor revisions, according to the following general and line-by-line comments:

**GENERAL COMMENTS:**

1. The interpretation of the factors controlling hillslope morphology (I suggest using "morphometry") is certainly supported by strong independent geological constraints on deep structures responsible to variable surface uplift along the transect used for the analyses. Nonetheless, this part of the discussions should clarify better the complete set of factors controlling the hillslope morphometry. For example, despite the climatic and lithological homogeneity, could the catchment size/shape play a role in the results obtained? Given that I'm convinced that high resolution DTMs are incredible sources of morphometric data to be interpreted in the light of the theoretical landscape evolution (Vergari et al., 2019, DOI: 10.1002/esp.4496), the high resolution itself imply that your DTM records very local surface features: despite the binning, how can you exclude that the results of your morphometric analyses along the transect are affected by the occurrence of local processes related to the hillslope-channel dynamics?

We do not observe any systematic trends or differences related to catchment size in the various plots, with smaller or larger catchments behaving as distinct groups. See for example the new figure 8 where symbol size is a function of catchment size (information about catchment area has been added to the caption of the figure). In most case, the calculations operated on the high resolution DTM are done by first fitting a quadratic surface over a circular neighborhood and then performing the operation of interest on this surface, as for example calculating the hilltop curvature (see Hurst et al., 2012 for example) (see Methods sections). In this case the size of the calculation window is 30 m, which has the effect of filtering the very local features mentioned.

2. Section 5.3 shows new data coming from the application of a dislocation model to predict surface deformations associated to tectonic structures. Therefore, I suggest moving it to the results, after having better explained the method you used.

Following a similar comment by reviewer #1 we have reorganized the presentation of this modeling aspect of the study, with a dedicated part in the *Methods*. However, for reasons developed above, we strictly restrict the *Data* section (previously *Results*) to the factual presentation of primary observations acquired in this study (topographic analysis and cosmogenic nuclides) and keep the modeling results in the *Discussion*, as these models are intended to support the interpretation and discussion of the data.

3. If I did not misunderstand, your inferences about the lacking record of the uplift gradient in the river long profiles is based on the analysis of all the catchments, so why do you show the profiles and CHI-plot just for a single catchment (fig. 5)? In my opinion this part should be better presented in order to strengthen the implications of your results, which concern the transient conditions recorded in the hillslope morphometry when the main trunks and tributary channels already adjusted to tectonic

perturbations (in case of low tectonic rates/relatively old tectonic input). I suggest you refer also to the papers by Demoulin (2011 doi:10.1016/j.geomorph.2010.10.033, 2012

doi:10.1029/2012GL052201), who based his landscape metrics on the diachronic response to tectonic perturbations by main trunks, tributaries and hillslopes within catchments.

Following similar comment by reviewer #1 we have reformulated the presentation of the results pertaining to river profile analysis. As stated above we have added a new figure showing that  $E^*$  and  $k_{sn}$  are significantly correlated. As explained above, our initial presentation of the  $k_{sn}$  data was ambiguous, as  $k_{sn}$  display variations along our transect (the northern basins display higher values as initially stated), but that there is an important scatter in the such data, making the analysis of the pattern difficult, whereas the much higher data density distribution of the hillslope metrics yield a much more resolved pattern. Citation to Demoulin (2012) added to the introduction.

4. In the discussions you state that "In the absence of any climatic, litho- logic or vegetation gradient, the observed increase in hilltop curvature, hillslope relief and normalized erosion rate points to a coincident increase in rock uplift". This seems to be a weakness of the method, since often such a homogeneity of climatic, lithologic and vegetation cover factors is lacking, especially when dealing with areas with regional extent. Maybe you'd better discuss this point before the conclusions.

Right, but such remark could be made for any type of geomorphological investigation over large areas. We have edited the discussion to better underline that this is a specificity of the studied area.

5. Some figure are too small to be readable (Figs 1, 6, 11).

The mentioned figures have been edited for better readability.

**LINE-BY-LINE COMMENTS:**

Title: I suggest changing "morphologic" with "morphometric"

We kept *morphologic* in the title, because *morphometric* was, in our sense, specifically referring to the measurement of the properties and was narrowing down the meaning much more than what we want to convey, whereas *morphologic* has a broader sense, referring to the general structure, evolution and behavior of the hillslopes.

L13: maybe you mean "eroded" conglomerate

No, the conglomeratic formation is the bedrock currently undergoing active erosion and providing the regolith mantling the hillslope

L26: types of forcing Done

L42-45: references for CHI maybe deserve to be cited

There is no explicit references to chi calculations here, but we have added more references in the methods.

L70: correct "hilltope" in "hilltop" Done

L129: Myr, not M.yr. Moreover, check the uppercase for East and West (uppercase not used in other sentences, please homogeneize) Done

L131: Lambruissier not readable in Fig. 1

**Done**

L151-153: The reason why the hillslopes can be considered as regolith-mantled and transport-limited systems is not so clear to me.

All our field observations show the existence of this mobile regolith cover on the hillslopes, such as the main limiting factor for hillslope evolution is the capacity to transport this material. This is now highlighted in the text.

L165: I'm not sure that "shallow" is the right adjective for slope. Done

L246: South to North, maybe? Done

L252-254: What stated is not so clear in Fig. 5: first of all because here are reported data only from a single catchment; secondly, not all the profiles shown have a concave-up shape (linear CHI-transformed shape).

Very small tributaries are not expected to collapse perfectly on the linear trend in chi-z space, due to change in dominant incision processes (increasing colluvial contribution for small channels), and some amount of dispersion around the main trend is usually observed. The main information here is the very good linearization of the main trunk profile. This information was added to the presentation of the data.

L256: South to North, maybe? Done (sentence actually modified as a response to rev #1 comment)

L265: Figure 9 is cited before Figure 8, maybe better inverting their numbering. Done

L265-266: The maximum denudation rate obtained from 26AI as declared in the text does not fit with the data plotted in Fig. 9B (here, sample P seems showing maximum denudation rate obtained from 26AI >88 mm/ka). Done (the range was actually excluding sample P)

L270: You should motivate the choice of considering only 10Be.

Done (lower uncertainty)

L279: what do you mean with "important" when referring to range in elevation of the catchments from South to North?

Done. Average elevation and relief of catchments indicated.

L307: Maybe "The uplift itself is due to a long. . ." Done. sentence modified for clarity

L324: E\* and CHT do not show a gradual increase as stated, rather an abrupt increase.

 $E^*$  is increasing from ~6 to its max value close to 10 in ~4 km, with a progressive evolution across this range, so we do not consider this change to be abrupt such as the one associated with a sudden offset. *Gradual* changed to *progressive*.

L341-341: maybe the lack of a correlation between LH and CHT differently from Hurst et al. 2013 could depend on the fact that in the submitted manuscript the metrics are measures along a narrow profile transverse to divides.

From the observation of Figure 1 of Hurst et al. (2013), it does not appear that  $L_H$  is varying as much as  $C_{HT}$  along their transect, and does not seem to follow the major

increase and then decrease in  $C_{HT}$  either. In Hurst et al. (2013) most of the studied hilltops are also oriented perpendicular to the transect.

L356: does not depend Done

L375: you'd better discuss the assumption of a single planar dislocation in an elastic medium used in tour modelling

Following similar comment by reviewer #1, all the presentation of the modelling approach has been moved to the Methods section (3.3), where the underlying assumption are now presented in more details.

L393: formations underwent long-wavelength Done

L413: . . . providing the framework

Done

L431: did you evaluate the possibility of other disturbances in your catchments before performing CRN analyses?

All the other sites were distinctively free of significant close disturbances. A benefit of sampling at hilltop sites instead at collecting sediments at outlet sites, following the widely used approach to derive Catchment-Wide Denudation Rates (CWDR) is that we are not concerned about the usual problems arising at the catchment scale (representativity of stochastic inputs, spatial variability of source material, variability of quartz content at the scale of the catchment, etc ...). Another benefit is that it allows to make a direct association between the denudation rate at the sampling site and the morphological properties measured in its direct neighborhood (instead of relying on spatially averaged metrics in the case of CWDR).

L478: maybe comparison instead of confrontation Done

**Reviewer #3 Peter van der Beek**

**General comments:**

Godard et al. exploit a high-resolution Lidar-derived digital terrain model to explore the hillslope response to a rock-uplift gradient in a well constrained, climatically and lithologically homogeneous setting: the Valensole Plateau in SE France. In particular, they show that systematic variations in ridge curvature and a derived nodimensional erosion rate can be linked to subtle tectonic uplift of the northern part of the plateau, which they proceed to model with an elastic-dislocation model as resulting from compressional reactivation of a pre-existing normal fault in the basement underlying the plateau. Finally, they use cosmogenic 10Be and 26Al measurements of ridge erosion rates to dimensionalise the analysis and provide estimates of the rates of shortening, uplift and erosion of the study area.

This manuscript will make an excellent contribution to ESurf. It shows how highresolution DTM data can be used in conjunction with cosmogenic nuclide data to provide detailed inferences about landscape response to variable uplift rates and shows a way forward in morphotectonic analysis. The analysis is very complete, as the authors assess both hill-slope and channel response, include cosmogenic nuclide data to obtain actual rates, and interpret the results in terms of a tectonic driver using a numerical model inversion. It is therefore timely and of broad interest to the ESurf readership. It is also (mostly) well written and illustrated. I therefore recommend this be accepted pending minor/moderate revision.

**Specific comments:**

There are a few issues the authors could address in more detail in a revision:

First, the uplift gradient discussed in the manuscript is superimposed on a longerwavelength uplift gradient that is clearly recorded in both present-day vertical motions as recorded by permanent GPS stations (e.g. Noguet et al., 2016), and deformed geomorphic surfaces such as the Valensole plateau and the alluvial terraces of the Durance River (e.g. Champagnac et al., 2008). While this is noted in passing in the manuscript, the tilting of the Valensole plateau would lead to increasing uplift/erosion rates toward the northeast even in the absence of an active fault. It could be made clearer in the manuscript that the authors are investigating a shorter-wavelength uplift pattern superimposed on the regional pattern. In keeping with this, if the interpretation of the authors is correct, the Durance river terraces, acting as passive markers, should also record the fault offset. Whereas the longwavelength tilting is recorded by these terraces (e.g. Champagnac et al., 2008 and references therein) it is not clear whether the faulting is. The terraces are reported to be up to 1 Ma old, i.e. up to about half the age of the surface of the Valensole plateau, and should therefore record up to half the offset of the surface (estimated at  $\sim$ 30 m from Figs. 2 and 10), largely sufficient to be recorded by the highresolution DTM. This aspect could be looked at, and the implications of the surface geometry of the terraces discussed, in some more detail.

Done. The differences between the short-wavelength and long-wavelength analysis are now underlined more clearly in the *Settings* and *Methods* sections.

The high resolution DTM inspection shows that actually the morphological expression of these Higher Terraces of the Durance (in particular HT2 from Dubar) as flat gentlydipping and low concavity surfaces is very limited due to the erosion of the plateau margin, and they are only present as very small discontinuous surfaces on some hilltops (which we excluded from our analysis), which are much smaller than what is mapped in Dubar (1984) or on the BRGM geological map (Fv). In particular these remnants are virtually absent from the northern part of the transect, making the detection of the finite deformation as recorded by the upper surface impossible. Along the Durance River, only the younger "Riss and Wurm" terrasses show preserved flat top surfaces. On its profiles, Dubar (1984) considers alluvial deposits of the "High Terraces" that do not show preserved terrace surfaces. The IGN 1m DEM clearly shows that, apart from localized and discontinuous remnants, there are no extensive flat top surfaces preserved in the studied area. Therefore, the HT1, HT2, HT3 and HT4 terraces of Dubar 1984, which correspond to the Fv and Fw formations on the BRGM geological map, cannot be used for accurate morphological analyses. The only surface that we could confidently use in this study is the Fu top surface of the plateau (which is plotted on figures 2 and 11). This has been indicated in the *Settings* section.

The analysis of the river profiles implicitly assumes they behave as detachmentlimited bedrock streams and that they have reached steady state with respect to uplift. Neither assumption is immediately obvious: given that these streams incise relatively non-consolidated coarse conglomerates, they could very well behave as transport-limited streams; and given the probably recent onset of faulting below the Valensole plateau, they may not have equilibrated to the uplift regime yet. Therefore, the fact that the stream-profile analysis led to less clear constraints than the hillslope analysis could be due to inadequate theory as much as insufficient resolution. Again, this aspect could be discussed in some more detail.

Done. The question of transport-limited conditions is now addressed in the discussion (5.2). We note that a prediction of such model would be some degree of sensitivity of concavity to rock uplift (e.g. Wickert & Schildgen, 2019) which is not observed with our dataset. Concerning a possible transient response, all rivers and tributaries display regular concave-up profiles, which collapse on single linear trends in chi-z space and so no evidence for such adjustment. At last, while the conglomerate weather progressively on hillsopes producing a continuous regolith cover, the initial un-weathered bedrock is highly resistant, as exemplified by the famous steep cliffs near Les Mées village, directly north of the studied transect. The corresponding information has been added to the *Settings*.

The discussion regarding whether the hill slopes have reached steady state (Fig 8) is not very clear. Some explanation of why R\* and E\* should vary as indicated in Fig. 8 in case of steady state appears to be lacking.

**Done. Explicit reference to equation 8 is now made in the text and the caption of the corresponding figure.**

Finally, the procedure to extrapolate locally measured erosion rates (Fig. 11) appears a little convoluted. I understand the authors' rationale for doing this, as it is not obvious in how much the measured erosion rates are representative, but the resulting pattern could be perceived as being somewhat removed from the actual data. A potential solution to this could be to simply plot the measured erosion rates at the corresponding locations in Fig. 11, so that it is easy to assess in how much the followed procedure modifies these measured rates.

Done. The corresponding figure has been modified accordingly. The overall increasing trend is preserved with the at-site denudation rates, with some scatter corresponding probably to short-wavelength variability in denudation rates. As indicated in the figure caption, these rates directly measured at the samples sites are systematically higher than those recalculated from spatially averaged curvature values. This is a direct consequence of a sampling bias toward high curvature ridges, which provided better conditions in the field. Assuming that equation 3 holds over the observed range of curvature (which should be the case if transport-limited conditions are maintained, as we observed at these sites) this should have no incidence on the calibration of the diffusion coefficient and hence on the computed continuous denudation transect (now in orange on the figure).

Technical corrections:

Although the writing is overall of good quality, a number of minor English-language issues distract somewhat from the science. These include frequent singular/plural confusions, illogical comma use and awkward sentence structures (placement of adjectives, etc.). I am returning an annotated manuscript directly to the authors so they can correct these minor issues.

**Done**

Figures are mostly clear and well drafted. I would suggest use of another colour than yellow for symbols or profile lines, as I found these very hard to read (e.g. in Fig. 2B, 8, 10A and B). Figure 2 A and C need a (larger) N-arrow to indicate they are rotated nearly 90  $\circ$ ; labelling the Durance River would also help.

**Done**

---

## Author Response (AR2)

Vincent GODARD
Associate Professor
Aix-Marseille Université
CEREGE
godard@cerege.fr

Aix-en-Provence, February 16th 2020

Pr. V. Vanacker
Editor
Earth Surface Dynamics

Dear Pr. Vanacker,

Please find below our response to your comments on the manuscript entitled *Hillslope denudation and morphologic response to a rock uplift gradient* (esurf-2019-50), on behalf of myself and co-authors.

Best regards,

Vincent Godard

Dear authors,

I have gone through the revised version of your paper, and appreciate the changes that were made in the text. Particularly, the changes made to the methods are useful, and improved the logical flow of the manuscript.

Before acceptation, there are several minor issues that need your attention.

(1) The methods now include the basics of the morphometric analyses, and the geomorphic transport laws that were used to derive the sediment fluxes (here, E*). You have added two equations at the beginning of the section (3.1.1.). I feel that a few sentences need to be inserted, so that the readers can easily grasp the logic flow of the section and the sequence of the equations. For example the link between the first two equations (Eq 1 and 2) and the remaining part of the section (Eq. 4) is not entirely clear to me. Also, note that you express in Eq. 2 Qs as a nonlinear function of the hillslope gradient, then argue for a linear relationship (L181), and then again use the Sc (critical hillslope gradient) in the remaining equations suggesting that you have non-linear behaviour. The text would be easier to follow if you insert a few lines of text between the equations. Also, check that you use systematically the same annotations. The combination of continuity equations (dQs/dx, expressed as a function of E) and z(x) is confusing.
Done. We have reworded the transition from equations 1&2 and equation 3. We have also added further clarification for a better understanding of the evolution from equation 1 to 4.

(2) The relationship between the denudation rates that were measured on the hilltops and the curvature is not straightforward. Figure 12A illustrates this issue, with a large scatter in the denudation rates for the range of hilltop curvatures. Even if you exclude the "P" sample, do you

have a significant relationship between the two variables? Which variable (beyond curvature) is explaining the increase in denudation rates that you observe along the transect? Can you elaborate on the variability of the denudation rates in the text of the discussion? Is there possibly a sampling bias (1 clast or amalgamation of clasts? shielding?)?

Indeed, one single value of $D$ can not account for all the measured $C_{HT}$ and $E$ at hilltop sites, with values spread from 0.003 to 0.006 m²/a. As discussed in the text, geological and climatic parameters are homogeneous over these sites, and we did not identify any systematic variations for a particular parameter that we could associate with these changes in $D.$ We have added a sentence concerning the eventuality of sampling bias. As discussed in the text, our favored interpretation for this range of values is the existence of authogenic transience for hillslopes leading to local acceleration or deceleration of erosion due to short-wavelength and short-lived changes in surface conditions. Such effects are usually averaged out when using catchment-wide denudation rates but are more visible in this study where we use direct measurements of denudation at hilltop locations.

Beyond these two comments, I have minor editorial comments

L134: What do you mean with "passive" benchmark.
Done. Changed to *passively deformed marker*

L134: tilting
Done

L135: I would delete "recent"
Done

L155-161: The argumentation on the transport-limited system is a bit confusing.
Done. The corresponding sentences have been reworded to better indicate that the first one was relevant to the hillslope dynamics, whereas the second one was dealing with the river network.

L208: hillslope
Done

L185: Can you clarify the definition of LH? Is LH, the maximum hillslope length?
Done. Horizontal distance from hilltop to channel

L191: In Eq. 7, isn't there a Er value missing in the right term?
No, E* is obtained by dividing equation 3 by equation 5. This is now explicitly indicated in the text

L225: Can you write the full equation? In the short version, you have written only dz in the left term, and dx in the right term.
Done

L228: In Eq. 12, you write z(x) and z(xb), while in previous equations, you have used the abbreviated form and have written "z". Please check for consistency.
Done

L288 and following: Can you include the units of the measures of hillslope gradients?
Done

L292: Why do you refer to "highly mobile conglomerate-derived class"? What is the size of the clasts? And what is the diffusive process responsible for mobilisation of the clasts along slope?

What we want to underline is the low cohesion of the regolith as an explanation for the low value of $S_c$. The information on clast size has been added to the settings. The main processes for downslope transport are creep and dry ravel. This is now indicated in the text, as well as a reference to field photograph 3B.

L306: The reference value of 0.25 is lower than what is commonly observed in literature for steady-state profiles? Typically, values are closer to 0.40 - 0.50. Why do you observe values of 0.25?
In the global compilation of Harel et al. m/n ranges from 0.16 to 0.63. One interesting features of their dataset is that high $K$ values (corresponding to highly erodible lithology such as the one we consider here) are often associated with low m/n values (below 0.3). So we do not consider the observed value to be particularly problematic. This information has been added to the text.

L329: Please rephrase "CHT undergoing a 2-fold increase from 7 to 10 km". Can you state from W to E?
Done

L367: Can you add the units here for Sc (m/m)?
Done

L368: What do you mean with "indicating possible decaying dynamics of the landscape"?
Done. "Toward lower relief added"

L369: appear
Done

L367-373: This section - where the authors argue that the R* vs. E* relationship could point to change in climatic boundary conditions along the transect - seems to be contradiction with the statements made on L328-338 of "homogeneity of geological, climatic and biological properties over the transect". Can you clarify your statement (L367-373)?
Done. An eventual change in climate would act at a regional scale and would still result in spatially homogeneous conditions at the scale of our transect.

L508-510: Here, you link the different denudation rates of the hilltop positions with the rock uplift rate patterns. How do you calculate the "implied slope rate of 20 mm/ka" from the differences in the denudation rates measured at the hilltop positions? What is the uncertainty on the derivation of the slip rates? Which assumptions are you using to derive slip rates based on surface denudation rates?
As indicated in the text the key assumption is that of topographic steady state, allowing to consider that denudation and uplift rates are equals. The text has been modified to include the various uncertainties.

L527: delete "globally"
Done

L529: Can you rephrase "between the two ends of the profile"?
Done. Changed to "along the profile"

Figure 2: In Figure 2, the authors map the travel times to the top of the Oxfordian marls. The link between the text (introduction to the study area) and the figure caption is not entirely clear. Can you introduce this information in the description of the study area? Can you explicitly mention why and how you use this information on the top of the basement in you study (for example L360 and following)?

Done. Presentation of the Mées Structure added to the settings section. In the discussion and the caption of figure 2 it has also been clarified that this information on basement geometry was derived from seismic surveys.